# Toward Cybersecurity-Expert Small Language Models

Matan Levi [1]  Daniel Ohayon [1]  Ariel Blobstein [1]  Ravid Sagi [1]  Ian Molloy [1]  Yair Allouche [1]

## Abstract

Large language models (LLMs) are transforming everyday applications, yet they lag behind in specialized fields, such as cybersecurity, due to a lack of high-quality, domain-specific models and training datasets. To address this gap, we present *CyberPal 2.0*, a family of cybersecurity-expert small language models (SLMs) ranging from 4B–20B parameters. To train CyberPal 2.0, we generate an enriched chain-of-thought cybersecurity instruction dataset built with our data enrichment and formatting pipeline, *SecKnowledge 2.0*, which integrates expert-in-the-loop steering of reasoning formats alongside LLM-driven multi-step grounding, yielding higher-fidelity, task-grounded reasoning traces for security tasks. Across diverse cybersecurity benchmarks, CyberPal 2.0 consistently outperforms its baselines and matches or surpasses various open and closed-source frontier models, while remaining a fraction of their size. On core threat-investigation tasks, such as correlating vulnerabilities and bug tickets with weaknesses, our best 20B-parameter model *outperforms GPT-4o, o1, o3-mini, and Sec-Gemini v1*, ranking *first*, while our smallest 4B-parameter model ranks *second*. On core cyber threat intelligence knowledge tasks, our models outperform almost all tested frontier models, ranking *second only to Sec-Gemini v1*. To foster reproducibility and practical adoption, we will release our models as open source.

## 1. Introduction

Threat management and security operations are a natural fit for language models, because they require reasoning over diverse evidence and security-specific context. One of the most promising directions for practical impact is threat management and security operations (Motlagh et al., 2024; Yao et al., 2024). This setting demands deep, cross-domain understanding spanning network, cloud, and application security, along with the ability to investigate and reason over threat reports (Zhang et al., 2025; Lin et al., 2025). In this paper, we focus on this direction and propose a single defensive model that supports the full security-operations loop. Our goal is a domain-specialized backbone that delivers core capabilities for detection, investigation, response, threat hunting, and data classification, while remaining straightforward to integrate and deploy within enterprise pipelines.

However, adopting frontier models for security is a challenging task. Commercial offerings typically enforce strict safety guardrails, which limit their practical utility in real-world security workflows (Weerawardhena et al., 2025). Additionally, full integration with organizational data sources is further constrained by compliance requirements, as security data often contains highly sensitive and private information (Zhang et al., 2025). For these reasons, many enterprises require on-premises solutions to meet privacy, compliance, and data residency obligations, making it impractical to send sensitive telemetry to external frontier services. Cost is another factor, where models must handle massive volumes of security data efficiently. These constraints make security another domain where domain-specific Small Language Models (SLMs) are preferable to general-purpose frontier models (Belcak et al., 2025).

In addition to practical deployability, such a model must support core capabilities in the cybersecurity domain. It requires deep technical grounding across multiple domains: operating systems, computer networks, cloud platforms, identity and access management, and enterprise security controls (Li & Liu, 2021; Aslan et al., 2023). Most importantly, the model needs to incorporate comprehensive understanding of threats that includes attacker tactics, techniques, and procedures (TTPs), as well as software vulnerabilities, weaknesses, adversary tooling, and probable attack paths. It must also align with security operation workflows, covering hypothesis-driven threat hunting, investigation, and severity assessment. Finally, such a model must connect the dots, reason effectively over security evidences, and deliver defensible conclusions that are accurate and reliable.

In earlier work, Levi et al. took an initial step in this direc-

---

[1]IBM Research. Correspondence to: Daniel Ohayon <daniel.ohayon@ibm.com>.

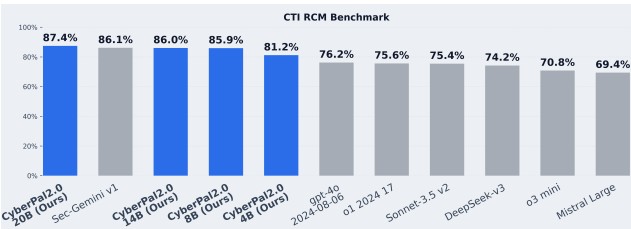

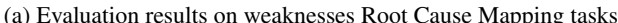

(a) Evaluation results on weaknesses Root Cause Mapping tasks.

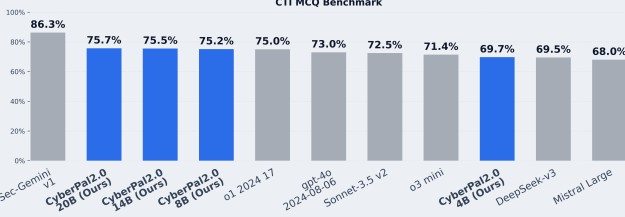

(b) Evaluation results on core cyber threat intelligence tasks.

*Figure 1.* Comparing our models (blue) against frontier models such as Sec-Gemini v1, o1, and o3-mini (gray) on key benchmarks.

tion. They introduced SecKnowledge, a domain-knowledge-driven cybersecurity instruction dataset built through a multi-phase generation process anchored in expert curation, along with SecKnowledge-Eval, a comprehensive evaluation suite covering a wide range of cybersecurity tasks. Models fine-tuned on SecKnowledge demonstrated significant improvements over baseline methods, highlighting the effectiveness of expert-guided instruction design and domain-specific evaluation in advancing cybersecurity LLMs.

This work takes a substantial step toward a practical security model for threat management and security operations:

- **SecKnowledge 2.0.** We propose a dataset enrichment pipeline that incorporates domain expertise via expert-in-the-loop schema-driven formatting, and applies multi-source, multi-step grounding to improve reasoning traces for security tasks and overall data quality.

- **Suite of cybersecurity-expert SLMs.** We train a suite of cybersecurity-focused SLMs, ranging from 4B to 20B parameters, that reason over complex threats and map domain knowledge to setting-specific analyses and recommendations.

- **Frontier cybersecurity performance.** Across rigorous cybersecurity benchmarks, our SLMs consistently outperform their baselines and state-of-the-art open-source models, yielding 7–14% average gains; on core cybersecurity threat intelligence (CTI) benchmarks, our models match or surpass frontier closed models (e.g., Sec-Gemini v1, OpenAI's o1), all while retaining the cost efficiency, openness, and on-premises deployability required by enterprises.

To support reproducibility and further research, we release the SecKnowledge 2.0 pipeline[1], CyberPal 2.0 20B model[2], and our evaluation benchmarks[3].

**Conflict of Interest Disclosure.** All authors are employed by IBM. The CyberPal 2.0 models introduced and evaluated in this paper were developed at IBM.

## 2. Related Work

Recent work positions LLMs as security tools across Cyber Threat Intelligence (CTI), malware analysis, and incident response, among other security tasks. Recent systematic reviews synthesize both the landscape and open gaps in evaluation and datasets (Zhang et al., 2025; Xu et al., 2024). In this work, we focus primarily on using LLMs as security tools and evaluate their performance in applied security settings.

Yu et al. curates a multi-source cybersecurity corpus for pre-training (web content, blogs, books, Wikipedia, and MITRE-linked resources), filters a general crawl for security-related text, and augments it with LLM-style rewrites; it then performs instruction fine-tuning on real-life cybersecurity-oriented tasks with LLM-generated references and distills reasoning on CTI-Bench using a general-purpose model with chain-of-thought. Despite the breadth of the pre-training data, their fine-tuning dataset is limited in size and is derived primarily via distillation. Following this work, Weerawardhena et al. created an instruction-tuned, security-specialized chat model built on the Foundation-Sec-8B base (a Llama-3.1-8B continued-pretrained on a curated cybersecurity corpus), which is competitive with both open and closed models like Gemma Team et al. and GPT-4o-mini (Hurst et al., 2024). The work minimizes security-specific content during post-training, relying on continued pre-training for domain knowledge; their post-training data emphasize diversity and instruction-following rather than security knowledge injection. Taken together, these studies emphasize security-focused pretraining, leaving underexplored the role of expert-driven, document-grounded supervised fine-tuning, which is crucial not only for reliability but also for enabling practical problem-solving and actionable guidance for cybersecurity workflows (Zhang et al., 2025).

Few practitioner-built checkpoints appear on community hubs (e.g., Hugging Face) without an accompanying paper or technical report (DeepHat-V1; SegoLily Labs). These releases often omit essential details (e.g., training data sources), making rigorous comparison and reproducibility challenging. Closed vendor security models sometimes like-

---

[1]SecKnowledge 2.0 code
[2]CyberPal 2.0 20B Model (HuggingFace)
[3]SecKnowledge-Eval benchmarks suite (HuggingFace)

wise report only headline scores. Google's Sec-Gemini v1 (Bursztein & Tishchenko, 2025) combines Gemini's reasoning with security knowledge and tools by tying in Google Threat Intelligence (Mandiant/GTI) and OSV[4]. They report strong results on CTI core knowledge and root-cause mapping tasks, though access remains limited.

We build upon SecKnowledge introduced by Levi et al., which takes a data-first route with an expert instruction set and an evaluation suite, and reported sizable improvements in threat-hunting Q&A and investigation assistance.

## 3. SecKnowledge 2.0: Data Reformatting and Enrichment Pipeline

In this section, we introduce *SecKnowledge 2.0* - an enhanced version of *SecKnowledge*, a comprehensive cybersecurity instruction dataset originally introduced by Levi et al., which generates synthetic data from curated cybersecurity seed sets. *SecKnowledge 2.0* extends *SecKnowledge* via a reformatting and enrichment pipeline, shown to improve downstream task performance (Fan et al., 2024; Nguyen et al., 2025; Abdin et al., 2024).

This section is organized as follows: Section 3.1 introduces *SecKnowledge*, which serves as our starting point dataset. Section 3.2 describes standard data reformatting and enrichment approaches. Section 3.3 then presents our improvements on top of the standard approaches described in 3.2, which combine LLMs with expert-in-the-loop feedback to define reasoning structures and employs LLM-automated query generation to retrieve external evidence for enriched, reliable responses. We use gpt-oss-120b with *Medium* reasoning effort as the backbone LLM. The result is *SecKnowledge 2.0*, a dataset whose responses are structured, interpretable, and supported by evidence.

### 3.1. SecKnowledge: A Diverse Set of Cybersecurity Instructions Set

**SecKnowledge** is a domain-knowledge–driven instruction dataset for cybersecurity, constructed in two stages that combine expert curation with structured automation. In the first stage, schema-based parsers were designed for foundational security corpora from public security data sources. In the second stage, SecKnowledge was then extended by generating high-quality seed instructions that capture both per data-source concepts and cross data-source relationships using a novel synthetic data generation method. For example, paths in BRON (a graph that interconnects security entities introduced in (Hemberg et al., 2020)) are transformed into chain-of-thought (CoT) (Wei et al., 2022) rationales. Sigma rules are converted into step-by-step "how to detect" ex-

planations, and SIEM rules are mapped to ATT&CK TTPs with grounded rationale. First stage yields ∼153k instructions across sources, providing a structurally diverse and practically grounded seed set. In the second stage, *SecKnowledge* increased both diversity and difficulty through dynamic content-grounded synthetic generation, yielding a 403k-example cyber-security corpus.

While *SecKnowledge* provides broad coverage and high-quality supervision, instruction families are intentionally template-based, which can yield limited rationales and shorter reasoning chains. Building on this foundation, our work enriches those items with explicit, step-by-step trajectories and stronger grounding by composing and adapting data reformatting and enrichment methods to the security domain, culminating in **SecKnowledge 2.0**.

### 3.2. Baseline: Data Reformatting and Enrichment Pipeline

We build on prior work showing that reformatting existing data sources, conditioned on chain-of-thought (CoT) reasoning, improves performance on downstream tasks (Fan et al., 2024; Nguyen et al., 2025; Abdin et al., 2024) and training token efficiency (Kimi Team et al., 2025). These pipelines often incorporate a stage that reformats raw answers into CoT reasoning traces, encouraging systematic reasoning. Within this line of work, Fan et al. introduces Reformatted Alignment (ReAlign), a format-driven pipeline that upgrades instruction datasets through three stages: (i) humans define CoT formats; (ii) enrichment adds auxiliary information; and (iii) reformatting imposes an explicit CoT structure.

Despite *SecKnowledge*'s breadth, responses tend to be concise with short rationales. A pipeline such as ReAlign can expand these compact answers into explicit, step-by-step trajectories while grounding them in retrieved documents and authoritative sources, making it a natural baseline and a strong foundation for *SecKnowledge 2.0*. At the same time, instruction generation in complex domains is prone to hallucinations (Jiang et al., 2023) and divergence from expert intent (Levi et al., 2025; Ramjee et al., 2025; Eachempati et al., 2025). We therefore believe a more domain-appropriate pipeline for cybersecurity should be adopted.

### 3.3. Pipeline Extensions

In the next section, we move beyond vanilla reformatting by introducing an expert-in-the-loop workflow that semi-automatically derives domain-specific formats for each task in the dataset. These formats specify the exact reasoning steps needed to reach the final answer. We further enhance them with document grounding and targeted web search, ensuring that each step is anchored in evidence and minimizing hallucinations during reformatting.

---

[4]https://www.mandiant.com/, https://osv.dev/

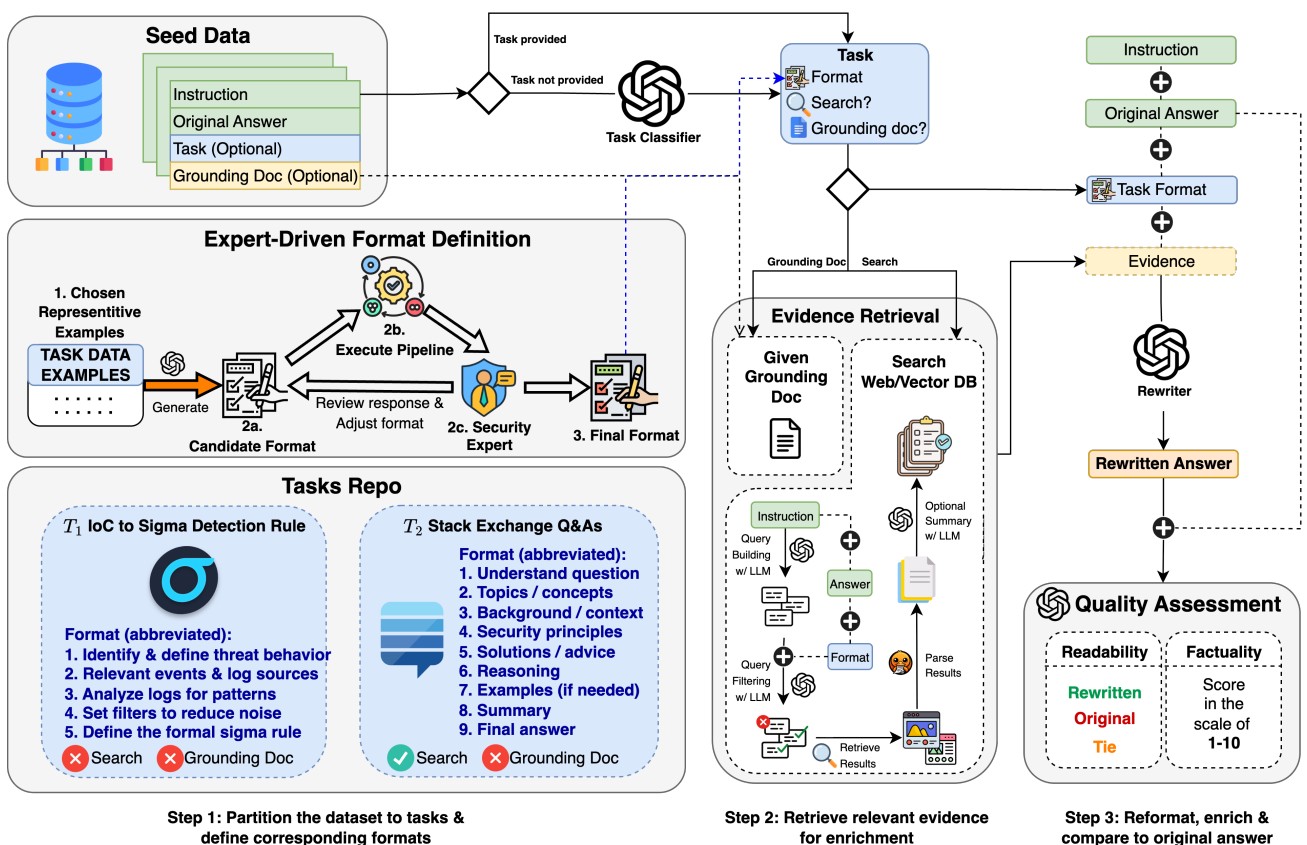

*Figure 2.* 3-step reformatting and enrichment pipeline overview - experts define task formats for the seed dataset, relevant evidence is retrieved from documents or web sources, answers are reformatted and enriched into structured, knowledge-grounded outputs with LLMaaJudge verification (see A.2).

### 3.3.1. EXPERT-IN-THE-LOOP SYSTEM FOR AUTOMATING DOMAIN-SPECIFIC FORMATS

For reformatting and enriching a given dataset $D$, the first step is to partition it into distinct tasks $\{T_1, ..., T_N\}$, each task representing a coherent sub-domain or capability. Formally: $S = \{T_1, ..., T_N | \bigcup_{k=1}^{k=N} T_k = D, T_i \cap T_j = \varnothing\}$. Since different problem types demand different ways of structuring outputs, each task $\mathbf{T_i}$ must be paired with a corresponding format $\mathbf{F_i}$ (refer to Figure 7 for an example format) that defines the task more precisely by specifying the steps needed to be taken to provide a detailed and logically coherent answer. Manually constructing such tailored formats, however, can be highly time-consuming, particularly in specialized domains such as cybersecurity, where expert knowledge is required, yet remains both scarce and costly.

To efficiently scale format definition across a large and hierarchical label space - such as SecKnowledge, which contains 105 unique tasks - we developed an expert-in-the-loop system capable of semi-automatically generating and evaluating format templates. The system employs a LLM that, given a concise task description together with an optional set of illustrative instruction–response examples from any task, generates a corresponding candidate output format. Within the same framework, experts can immediately evaluate this format by executing the full pipeline on representative inputs, obtaining rewritten responses along with auxiliary feedback such as search results and LLM-as-a-judge scores for readability and factuality. Based on this feedback, experts can directly edit the format and rerun the pipeline, enabling a tight feedback loop that supports iterative refinement while substantially reducing the manual burden of format specification and enhancing the efficiency, accuracy, and scalability of the pipeline. For implementation details, refer to Appendix A.1.

### 3.3.2. LLM-GUIDED SEARCH AND DOCUMENT GROUNDING PIPELINES

The vast majority of tasks in SecKnowledge can greatly benefit from enrichment through evidence retrieval. Such grounding is necessary to reduce the risk of LLM hallucinations when rewriting responses and to ensure that outputs remain accurate and reliable. Evidence can be provided

in two primary ways: (1) by attaching a grounding document directly to the instruction–response pair, such as an advanced persistent threat (APT) report describing a specific attack, or (2) by searching for relevant documents on demand. The first method is largely straightforward, as it simply links an instruction–response pair to the document from which it was derived. The second method, however, requires more substantive mechanisms, such as searching a pre-indexed corpus (e.g., via a vector database) or the world wide web. Accordingly, the discussion that follows concentrates on the latter, given its broader applicability and scalability. To obtain high-quality search results, we design the mechanism as a structured multi-step process:

1. **Query building.** Given only the instruction, the LLM is prompted to generate $K$ candidate search queries. This stage can be viewed as a brainstorming step to provide diverse queries.

2. **Query filtering.** A second LLM, conditioned on the instruction, the original answer, the task format, and the candidate queries, selects only those queries expected to provide new or useful information that can fill gaps.

3. **Results retrieval.** The filtered queries are then executed against either a vector database or the web, with the top $R_{max}$ results retrieved for each query (yielding $\leq K \times R_{max}$ results).

4. **Results parsing.** For each query, the top $R$ results that could be parsed to text are retained, while unprocessable ones (e.g., websites that block automated access) are discarded.

5. **Optional summarization.** When retrieved documents are large (e.g., web pages), they may be summarized in a manner conditioned on the task format, ensuring that the retained content aligns with the information required to populate the format steps.

In our experiments, we used $K = 2$, $R_{max} = 8$, and $R = 2$, without applying summarization. This configuration is motivated by two considerations: (1) automatic summarization often omits critical details required by the task format, and (2) full documents, when not summarized, can accumulate into a large number of tokens that risk exceeding the context window allocated for the LLM. Moreover, even when the context window is not exceeded, long inputs can lead the model to disproportionately attend to the beginning and end of the prompt rather than its middle content (Liu et al., 2023), effectively nullifying the benefit of providing additional context. To balance these constraints, we restricted retrieval to at most four results ($K \times R = 4$), thereby prioritizing fewer, higher-quality search results over larger volumes of noisier content.

## 4. Training and Evaluation Process

We detail the training and evaluation pipelines. For the evaluation, we report results on widely used open-source security benchmarks. When comparing to state-of-the-art frontier models (such as Sec-Gemini v1), we follow the evaluation protocol of Sec-Gemini v1 and report results on the well-known CTIBench-RCM and CTIBench-MCQ (Alam et al., 2024). Additionally, for both the baseline and our fine-tuned models, we evaluate on further benchmarks (Li et al., 2023; Tihanyi et al., 2024; Levi et al., 2025), as elaborated in Section 4.2.

### 4.1. Training Recipe

To train our models, we use our generated *SecKnowledge 2.0* dataset. We employ Qwen3-4B-base, Qwen3-8B-base, and Qwen3-14B-base, alongside gpt-oss-20b as our starting point. Training is performed with a learning rate of $4 \times 10^{-5}$ and a linear warm-up ratio of 0.15. The context length is set to 8192, and the batch size is 3072. We train our models for two epochs.

To train our models with adaptive reasoning capabilities, we incorporate adaptable reasoning depth: long-form chain-of-thought examples from SecKnowledge 2.0 are augmented with "step-by-step" requests, while shorter instructions from the original SecKnowledge dataset are paired with concise, fast-response requests. This design, similar to the notion of reasoning effort in gpt-oss, balances reasoning-intensive and lightweight tasks. For the shorter, fast-response requests, we sample approximately 25% of the original instructions and responses from the original SecKnowledge dataset, focusing primarily on short, high-quality examples selected using LLMaaJ. The procedure of mixing a portion of the high-quality original responses with their enhanced counterparts not only teaches the models to perform adaptive reasoning, but also improves token utility by amplifying the volume of high-quality tokens while reducing overfitting, as observed by Kimi Team et al. (2025).

Additionally, we observed a phenomenon also reported in recent studies (Huerta-Enochian & Ko, 2024; Shi et al., 2024; Chatterjee et al., 2025): it is often preferable to retain at least partial loss on the prompt rather than masking it out entirely during training. Finally, we conducted experiments to determine whether a base or a post-trained model is a better starting point for fine-tuning. We found that base models tend to learn more effectively than their post-trained counterparts. Appendix B presents additional training details, alongside a small-scale experiment comparing Qwen3-8B and Qwen3-8B-post-trained under the same training recipe, illustrating the differences between starting from a base versus a post-trained model.

**Instruction**
What are the required skills to execute the attack pattern: Leveraging Active Adversary in the Middle Attacks to Bypass Same Origin Policy (CAPEC ID: 466)

**SecKnowledge Answer**
The skills required to execute Leveraging Active Adversary in the Middle Attacks to Bypass Same Origin Policy (CAPEC ID: 466) are the following:
1. Ability to intercept and modify requests / responses The level of the required skills can be classified as Low.
2. Ability to create iFrame and JavaScript that would initiate unauthorized requests to sensitive sites from the victim's browser The level of the required skills can be classified as Medium.
3. Solid understanding of the HTTP protocol The level of the required skills can be classified as Medium

**SecKnowledge 2.0 Answer**

**\*\*1. Attack Pattern Overview\*\***
*CAPEC-466 – Leveraging Active Adversary in the Middle Attacks to Bypass Same Origin Policy*
This pattern describes how an attacker positioned as a man-in-the-middle (e.g., on a public Wi-Fi) injects an invisible iFrame into an unencrypted HTTP response. The iFrame forces the victim's browser to issue automatic requests to a privileged domain, which the attacker then intercepts, redirects, and enriches with malicious JavaScript. By exploiting the Same Origin Policy, the attacker can steal cookies (including HTTP-only), credentials, and other sensitive data from any site they choose.

**\*\*2. Technical Mechanics Involved\*\***
- Interception and modification of HTTP requests/responses on an unsecured network.
- Injection of iFrames that trigger cross-origin requests from the victim's browser.
- Real-time redirection of those automatic requests back to attacker-controlled services.
- Injection of malicious JavaScript into the responses of the targeted privileged site, allowing code execution under that site's origin.
- Exploitation of persistent and session cookies without needing XSS-style vulnerabilities.

**\*\*3. Core Skills Required\*\***

| Skill | Definition & Relevance | Difficulty to Master | Impact on Attack Effectiveness |
|---|---|---|---|
| **\*\*Intercept & modify HTTP traffic\*\*** | Ability to capture, read, and alter HTTP requests and responses flowing between the victim and web servers (e.g., using proxy tools or packet sniffers). This is essential for inserting the malicious iFrame and for redirecting the victim's automatic requests. | \*\*Low\*\* – Basic networking tools (Wireshark, mitmproxy) suffice; no deep programming needed. | Enables the initial foothold; without it the attacker cannot inject the iFrame or reroute traffic, making the attack impossible. |
| **\*\*Create iFrames and craft JavaScript for unauthorized cross-origin requests\*\*** | Skill in writing HTML iFrames and JavaScript that silently cause the browser to send requests to a target domain, then capture or manipulate the responses. This drives the victim's browser to interact with the privileged site under the attacker's control. | \*\*Medium\*\* – Requires familiarity with client-side scripting, DOM manipulation, and browser security models. | Determines how stealthy and reliable the request redirection is; well-crafted scripts reduce detection and increase success rates. |
| **\*\*Solid understanding of the HTTP protocol\*\*** | Comprehensive knowledge of HTTP methods, headers, cookies, and the mechanics of request/response cycles. Critical for correctly modifying traffic, preserving session cookies, and ensuring the injected content is accepted by the browser and server. | \*\*Medium\*\* – Involves grasping protocol nuances, status codes, and security headers. | Directly affects the precision of traffic manipulation; errors can break the session or alert the victim, lowering attack reliability. |

**\*\*4. Conclusion\*\***
Executing CAPEC-466 hinges on three primary capabilities: low-level traffic interception, intermediate client-side scripting to embed malicious iFrames, and a solid grasp of HTTP fundamentals. While the interception skill is relatively easy to acquire, the scripting and protocol expertise are moderately demanding but crucial for maintaining stealth and ensuring the attack's effectiveness. Mastery of these areas enables an adversary to bypass Same Origin Policy protections and harvest sensitive data from targeted sites.

*Figure 3.* Q&A example from SecKnowledge (green), and our improved answer (orange).

## 4.2. Evaluation benchmarks

We evaluate models exclusively on cybersecurity benchmarks spanning governance/compliance, architecture/operations, and threat detection & response. The suite emphasizes document-grounded reasoning, consistent mapping across security taxonomies, and resilience to adversarial distractors. Our evaluation uses the benchmarks listed below; See Appendix C.1 for further details.

**CTI-MCQ** tests breadth of CTI knowledge via multiple-choice items on attack patterns, actors, detections, mitigations, and frameworks (Alam et al., 2024). **CTI-RCM** evaluates document-grounded root-cause mapping by linking CVE evidence and bug reports to the correct CWE(s) with taxonomy-aware disambiguation (Alam et al., 2024). **SecEval** offers over 2,000 multi-option questions across nine security domains from authoritative sources, measuring accurate recall and the ability to apply controls and frameworks to concrete scenarios (Li et al., 2023). **CyberMetric-2000** comprises 2000 expert-validated questions spanning diverse subdomains, indicating professional-level declarative security knowledge under closed-book conditions (Tihanyi et al., 2024). **CISSP exams** contains questions drawn from the assessment tests within the CISSP learning material, assessing analysts' skills across the entire security posture. **Technical Weakness Impact Mapping** requires assigning CWE descriptions a weakness to one or more of eight technical impacts, emphasizing consequence-centric reasoning

beyond exploitability (Levi et al., 2025). **Adversarial CTI** ties questions to specific MITRE ATT&CK entities and uses adversarial distractors to probe robustness on campaigns, tactics, detections, and mitigations (Levi et al., 2025). **CTI Detection & Mitigation** checks whether models propose appropriate detections and mitigations for tactics/techniques, attack patterns, weaknesses, and vulnerabilities (Levi et al., 2025). **CTI Relationship Prediction** tests cross-taxonomy reasoning and relationship *hallucinations* by choosing the correct justification for whether two CTI entities (e.g., CVE and CWE mapping) are related (Levi et al., 2025).

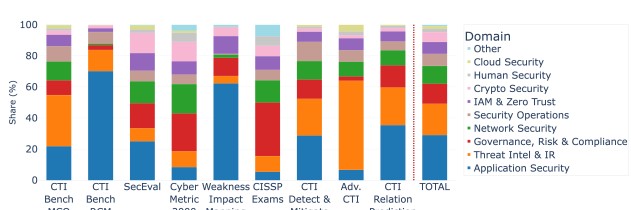

*Figure 4.* Domain composition of each data source in the evaluation benchmarks.

Figure 4 shows the benchmark's distribution across the cybersecurity taxonomy used to quantify coverage (see Appendix C.2.1). As can be seen, the benchmarks are closely aligned with organizational security priorities, with a particular focus on *threat Intelligence*, *incident response*, *security operations*, *application security*, and *identity management*.

### 4.3. Evaluation Process and Metrics

Similar to (Wang et al., 2024), we also found that models utilizing Chain of Thought (CoT) reasoning achieved better performance on complex security benchmarks, compared to direct answering. Therefore, we utilize a zero-shot CoT prompting. The CoT template incorporates essential reasoning steps and format to allow models to easily follow the given instructions. We used zero-temperature for consistency. We then use a regular expression parser to extract the final answer from the model's CoT process. The prompt used in the evaluation is provided in Figure 10 in Appendix D. For the Qwen suite of models, we compared our fine-tuned versions to baseline models (post-trained) with the *thinking* flag enabled, allowing them to leverage their reasoning process. For gpt-oss, we used reasoning effort *Medium* to avoid failures caused by the full CoT exceeding maximum window size.

## 5. CyberPal 2.0: A Suite of Cybersecurity Language Models

To demonstrate the effectiveness of our method, we train a family of security-expert SLMs ranging from 4B to 20B parameters. We then report results against the post-trained versions of the same base models from which our models were fine-tuned, alongside results against state-of-the-art frontier models (e.g., o1, Sec-Gemini v1). Lastly, we perform ablation studies.

### 5.1. Results

**Results Compared to Baselines.** Table 1 presents results compared to the baseline models from the same families that CyberPal 2.0 was fine-tuned from. Meaning, for example, we measure improvements across various benchmarks between Qwen3-8B and CyberPal 2.0-8B, which was fine-tuned from Qwen3-8B-base. On average, our models outperform their baselines by 7–14%. We also observe substantial gains on key benchmarks such as CTIBench-RCM, where our models exceed the baselines by 16–31%, and CTIBench-MCQ, where they achieve improvements of 8–12%. We also include gpt-oss-120B as a reference to highlight our models' strong performance.

**Results Compared to Open Source cybersecurity models & Additional Open Source Model Families.** We evaluated our models against recent 7B–8B open-source cybersecurity models; our 8B leads across all benchmarks. Full details are in Appendix E. Additionally, in Appendix F we evaluate our models against other open source model families.

**Results Compared to Frontier LLMs.** We evaluated our models against state-of-the-art general models such as Sec-Gemini v1 and OpenAI's o1. As evaluation is costly, we

follow Sec-Gemini v1 evaluation protocol and report results on the CTIBench benchmarks: CTIBench-MCQ and CTIBench-RCM (Alam et al., 2024). CTIBench-MCQ assesses an LLM's understanding of core cyber–threat intelligence concepts, while CTIBench-RCM evaluates model's ability to perform Root Cause Mapping (RCM), identifying the underlying causes of vulnerabilities by correlating vulnerabilities records and bug tickets with weaknesses. This benchmark is considered a leading threat intelligence benchmark and serves as a strong indicator of a model's threat management capabilities.

As shown in Figure 1, our models are on par with—or better than—most frontier models. **RCM:** CyberPal 2.0–20B ranks first overall, surpassing Sec-Gemini v1; the 14B, 8B, and 4B variants all ranked second and exceed the remaining frontier models. **MCQ:** the 20B and 14B models ranks second and third respectively, immediately behind Sec-Gemini v1 and ahead of o1; the 8B model is competitive with GPT-4o; and even the 4B model outperforms much larger models such as Mistral Large and DeepSeek-v3, with performance close to o3-mini.

### 5.2. Results on Real-World Use Cases

Beyond standard cybersecurity benchmarks, we evaluate CyberPal 2.0 on two applied workflows: mapping threat reports to MITRE ATT&CK techniques and reassessing CVE severity under product-specific constraints. These tasks test whether the models generalize to realistic security artifacts and operational decision-making settings. Additional real-world evaluations, including CyberSOCEval and secure code generation, are reported in Appendix J.

**Threat reports to TTP mapping.** We construct this benchmark from MITRE campaign entries and their references. Each campaign lists techniques used by the adversary, and some technique entries cite APT reports describing the use of that technique. We use these reports to build a multiple-choice task in which the model receives an unstructured threat report and selects the corresponding ATT&CK technique. This evaluates whether the model can connect realistic CTI evidence to operational security taxonomies.

**CVE reassessment.** We also evaluate CVE reassessment, where the model must reassess a CVSS[5] score for a specific package, product, or deployment context. This setting tests whether the model can adjust generic vulnerability severity according to concrete deployment constraints such as packaging, exposure, configuration, and permissions. We report normalized Mean Average Distance, where higher is better.

Table 2 shows consistent gains across both settings. On threat-report mapping, CyberPal 2.0 improves over all cor-

---

[5]https://www.first.org/cvss/v3.1/specification-document

*Table 1.* Evaluation results for CyberPal 2.0 models compared to their corresponding baseline (post-trained) models and the gpt-oss-120B open-source model.

| Model | CTI Bench MCQ | CTI Bench RCM | SecEval | Cyber Metric 2000 | CISSP Exams | Adv. CTI | Weakness Impact Mapping | CTI Detect & Mitigate | CTI Relationship Prediction | Avg. |
|---|---|---|---|---|---|---|---|---|---|---|
| Qwen3-4B | 61.88 | 49.95 | 57.38 | 87.40 | 79.80 | 64.51 | 57.02 | 60.77 | 67.99 | 65.19 |
| **CyberPal-2.0-4B** | **69.70** | **81.15** | **59.02** | **87.80** | **80.80** | **68.03** | **66.48** | **64.03** | **77.12** | **72.68** |
|  | (+7.82) | (+31.20) | (+1.64) | (+0.40) | (+1.00) | (+3.52) | (+9.46) | (+3.26) | (+9.13) | (+7.49) |
| Qwen3-8B | 63.13 | 63.25 | 56.19 | 88.45 | 83.33 | 64.93 | 53.58 | 59.88 | 60.67 | 65.93 |
| **CyberPal-2.0-8B** | **75.15** | **85.95** | **66.93** | **89.85** | **88.89** | **87.61** | **71.06** | **70.26** | **87.66** | **80.37** |
|  | (+12.02) | (+22.70) | (+10.74) | (+1.40) | (+5.56) | (+22.68) | (+17.48) | (+10.38) | (+26.99) | (+14.44) |
| Qwen3-14B | 64.28 | 70.50 | 61.48 | 89.85 | 86.36 | 69.43 | 62.46 | 63.44 | 58.48 | 69.59 |
| **CyberPal-2.0-14B** | **75.51** | **86.00** | **69.71** | **89.95** | **90.40** | **89.58** | **70.77** | **70.95** | **92.93** | **81.76** |
|  | (+11.23) | (+15.50) | (+8.23) | (+0.10) | (+4.04) | (+20.15) | (+8.31) | (+7.51) | (+34.45) | (+12.17) |
| gpt-oss-20B | 64.57 | 68.95 | 67.65 | **90.20** | 79.80 | 61.83 | **71.91** | 67.49 | 65.42 | 70.87 |
| **CyberPal-2.0-20B** | **75.71** | **87.40** | **72.86** | 89.05 | **86.87** | **84.93** | 70.77 | **67.69** | **87.66** | **80.33** |
|  | (+11.14) | (+18.45) | (+5.21) |  | (+7.07) | (+23.10) |  | (+0.20) | (+22.24) | (+9.46) |
| gpt-oss-120B | 69.37 | 79.95 | 68.02 | 92.55 | 84.34 | 72.76 | 65.90 | 64.52 | 70.56 | 74.21 |

*Table 2.* Real-world use-case results.

| Model | Threat→TTP (Accuracy) | CVE Reassess. (Norm. MAD) |
|---|---|---|
| Qwen3-4B | 74.00 | 0.783 |
| **CyberPal 2.0-4B** | **82.75** | **0.830** |
|  | (+8.75) | (+0.047) |
| Qwen3-8B | 74.00 | 0.825 |
| **CyberPal 2.0-8B** | **78.25** | **0.834** |
|  | (+4.25) | (+0.009) |
| Qwen3-14B | 79.25 | 0.740 |
| **CyberPal 2.0-14B** | **85.25** | **0.833** |
|  | (+6.00) | (+0.093) |
| gpt-oss-20B | 76.75 | 0.738 |
| **CyberPal 2.0-20B** | **79.50** | **0.834** |
|  | (+2.75) | (+0.096) |

responding baselines: all models gain 2.75–8.75 points. On CVE reassessment, all CyberPal 2.0 variants improve over their baselines, with the largest gains for the 14B and 20B models. These results suggest that SecKnowledge 2.0 improves practical security reasoning beyond standard benchmark formats.

### 5.3. Ablation Studies

We conduct ablation studies to identify which components of SecKnowledge 2.0 are responsible for the observed gains. In particular, we study three factors: (1) the data pipeline used to construct the training set, (2) the contribution of

retrieval grounding through the search component, and (3) the choice of reformatting backbone LLM. All ablations are run with the same target model (Qwen3-4B-Base), training budget, configuration, optimization settings, and context settings.(as described in §4.1) The ablation were further evaluated on all nine benchmarks. Table 3 summarizes the average accuracy across all ablations, and Appendix G reports the full benchmark-level results.

**Effect of the data pipeline.** We first compare models trained on the original SecKnowledge dataset with models trained on data produced by a reformatting baseline inspired by Fan et al. (2024), and with models trained on data produced by our full SecKnowledge 2.0 pipeline. Specifically, in the baseline setting we keep our pipeline fixed and rewrite the original SecKnowledge dataset using general-purpose question-answering formats instead of expert-written task-specific formats. Training on the original SecKnowledge already improves over the base model, confirming the value of domain-specific cybersecurity supervision. Applying generic reformatting yields further gains, but our full SecKnowledge 2.0 pipeline performs best overall. This indicates that the improvements are not explained by reformatting alone, but by using expert-defined task structures and enriched supervision tailored to the cybersecurity domain.

**Effect of retrieval grounding.** We next assess the contribution of the search component by comparing the full pipeline with a variant in which search is removed. Removing search reduced the average performance, showing

*Table 3.* Summary of ablations on the Qwen3-4B-base setting. We report average accuracy across all nine benchmarks.

| Data Pipeline | Formats | Backbone LLM | Search | Avg. |
|---|---|---|---|---|
| Qwen3-4B base | – | – | – | 65.19 |
| SecKnowledge (Levi et al., 2025) | – | – | – | 68.60 (+3.41) |
| SecKnowledge 2.0 (Ours) | General-purpose (Fan et al. (2024)) | gpt-oss-120B | ✓ | 70.04 (+4.85) |
| SecKnowledge 2.0 (Ours) | Expert-written | gpt-oss-120B | ✗ | 71.11 (+5.92) |
| SecKnowledge 2.0 (Ours) | Expert-written | Llama-4-Maverick | ✓ | 72.32 (+7.13) |
| SecKnowledge 2.0 (Ours) | Expert-written | gpt-oss-120B | ✓ | **72.68 (+7.49)** |

that retrieval-based grounding improves the quality of the generated supervision. This result is consistent with our hypothesis that external evidence helps reduce hallucinations and produces more reliable rewritten answers.

**Effect of the reformatting backbone LLM.** Finally, we examine the effect of replacing the reformatting backbone LLM from gpt-oss-120B with Llama-4-Maverick (Meta AI, 2025) while keeping the rest of the pipeline unchanged. Despite the large difference in scale between the two reformatters (120B vs. 400B parameters), the resulting average performance is nearly unchanged. This suggests that the gains are not tied to a specific large teacher model, and that a different reformatter can support almost the same downstream performance within our pipeline.

### 5.4. LLM-as-a-Judge Evaluation

Finally, we assess answer quality via *LLM-as-a-Judge* (LLMaaJ) (Zheng et al., 2023). Thirty cybersecurity experts authored 115 open-ended questions spanning command-line risk assessment, enterprise security, general cybersecurity, network security, and CTI-related topics. For each question and pair of model answers, the judge receives expert-curated grounding documents. We use OpenAI's o3 as the judge.

We evaluate answer quality using pairwise LLM-as-a-Judge comparisons. To validate the judge, we measure its agreement with human cybersecurity experts. We find that o3 agrees with human preferences in 80% of cases without grounding, and in over 90% of cases when provided with carefully collected grounding documents. This suggests that grounded o3 judgments are a reliable proxy for human preference in our setting.

Using the previously constructed open-ended cybersecurity questions, we generate answers from different model variants and ask the judge to compare each pair of responses.

The judge receives the question, two candidate answers, and the relevant grounding documents, and decides which answer is better. Figure 5 shows that CyberPal-2.0 is consistently preferred over both baselines. Further details on the question set, evaluation protocol, and additional per-category results are provided in Appendix H.

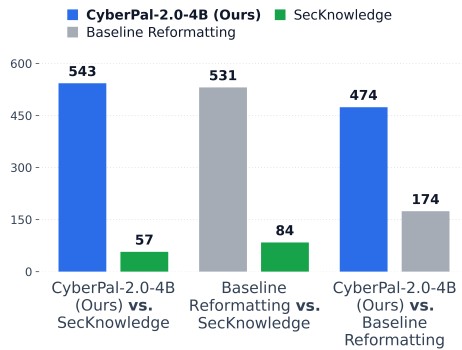

*Figure 5.* Pairwise comparison results from LLMaaJ (o3) with *grounding*, aligned with human experts.

## 6. Conclusion

**CyberPal 2.0** demonstrates that compact, domain-specific SLMs (4B–20B) can deliver frontier-level capability for security operations without frontier-level cost. Built on *SecKnowledge 2.0*—with schema-driven reformatting, expert-in-the-loop enrichment, and a multi-step grounding process—our models achieve 7–14% average gains over strong open-source baselines and, on core CTI tasks, match or surpass leading closed models; notably, the 20B model outperforms GPT-4o, o1, o3-mini, and Sec-Gemini v1, while even the 4B variant ranks second. Ablations and LLMaaJ validated by human experts attribute the gains primarily to data-quality improvements from our **SecKnowledge 2.0** enhanced reformatting and enrichment pipeline.

## Impact Statement

While our training focused on defensive threat management and security operations (e.g., threat investigation and incident response), the models' enhanced security knowledge could be misused by malicious actors in unforeseen ways. To reduce misuse risk, we (i) rely mostly on publicly available, open-access, non-sensitive sources; (ii) avoid training or releasing offensive cybersecurity capabilities; and (iii) will distribute models under responsible-use terms with safety filters and red-teaming.

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

## A. Pipeline Extensions Details

### A.1. Format Generation Framework

The system is composed of three primary stages, with the second and third stages forming an iterative loop that can be repeated until the user is satisfied with the resulting format, as illustrated in Figure 6. It is important to note that this framework is applicable only once the set of tasks covering the entire dataset has been defined, at which point the process is limited to the generation of formats.

**Data Exploration & Example Selection.** First, the user selects the appropriate category and task from the menus. From $N$ available examples for each task ($N = 500$ in our case), the system samples $k$ instruction–answer pairs for inspection ($k = 1$ in Fig. 6). This stage enables the user to explore the dataset - examining the range of questions and corresponding answers for the selected task - and to identify representative examples. These examples are subsequently used both to guide format generation and as inputs to the pipeline.

**Format Generation.** Second, the user provides a brief description of the task, this description will be used both to classify unlabeled instructions from the dataset and to generate the format. Then he selects an LLM to produce a candidate format using one of the available prompts. In our case, two distinct prompts were required: one tailored for specific tasks - such as the instructions generated in the first stage of *SecKnowledge*, which originate from a defined source and consistently ask for the same type of information, albeit in different contexts - and another designed for more general tasks, which encompass a wide variety of instructions, as in the second stage of *SecKnowledge*. In the latter case, providing examples may bias the format toward the selected instances, which is undesirable. The framework further supports the seamless addition of new prompts if needed. Once generated, the format can be refined by the user through manual editing.

**Evaluation Through Pipeline Execution.** Third, the user can run the pipeline on any example, with the first example automatically pre-filled by default. During this step, the user may adjust various hyper-parameters - for instance, enabling or disabling web search, specifying the number of search queries and the number of results per query, and deciding whether to summarize each retrieved before including it in the rewriting context. A grounding document can also be provided, either as an alternative to or in addition to web search. The pipeline then outputs the rewritten response, quality assessment scores, and, if requested, the retrieved search results.

### A.2. Data Generation Quality Assessment

After rewriting the original answer according to the format, the pipeline also incorporates evaluation in the form of LLM as a Judge. There are 2 criteria by which we judge the answers generated by the pipeline.

1. **Readability.** We prompt the judge with the instruction, and both answers and ask it to select the better answer according to the criteria described in Figure 11 (original and rewritten). We do this two times - in the first time the original answer is first and the rewritten is second, and in the second time the order is the opposite (while anonymizing which one is the original one and which one is the rewritten one). We test both directions to avoid positional bias (Wang et al., 2023; Zheng et al., 2023).

2. **Factuality.** We prompt the judge with the original answer and the rewritten answer, emphasizing that the original answer is the ground truth, and ask the LLM to provide a score in a scale of 1-10 that determines how factual the rewritten answer with respect to the original answer.

By combining these two criteria, we can get a sense of the quality of the defined formats and the new dataset. In Figure 8, we present the quality assessment results on our dataset, *SecKnowledge 2.0*. On average, the new answers are preferred 85.62% of the time (with 5.55% to the favor of the original answers, and 8.56% of inconsistency, where switching the position of the answers changed the judge decision, the rest are ties), while maintaining the factuality, reflected by an average factuality score of 9.25. These results show that our pipeline is robust.

## B. Training Recipe Additional Details

**Training from base vs. post-trained model.** In Table 4, we present a small-scale experiment examining the effect of the starting checkpoint *using an identical training recipe and evaluation protocol*. Using Qwen3-8B, we evaluate how

---

[6]https://www.gradio.app/

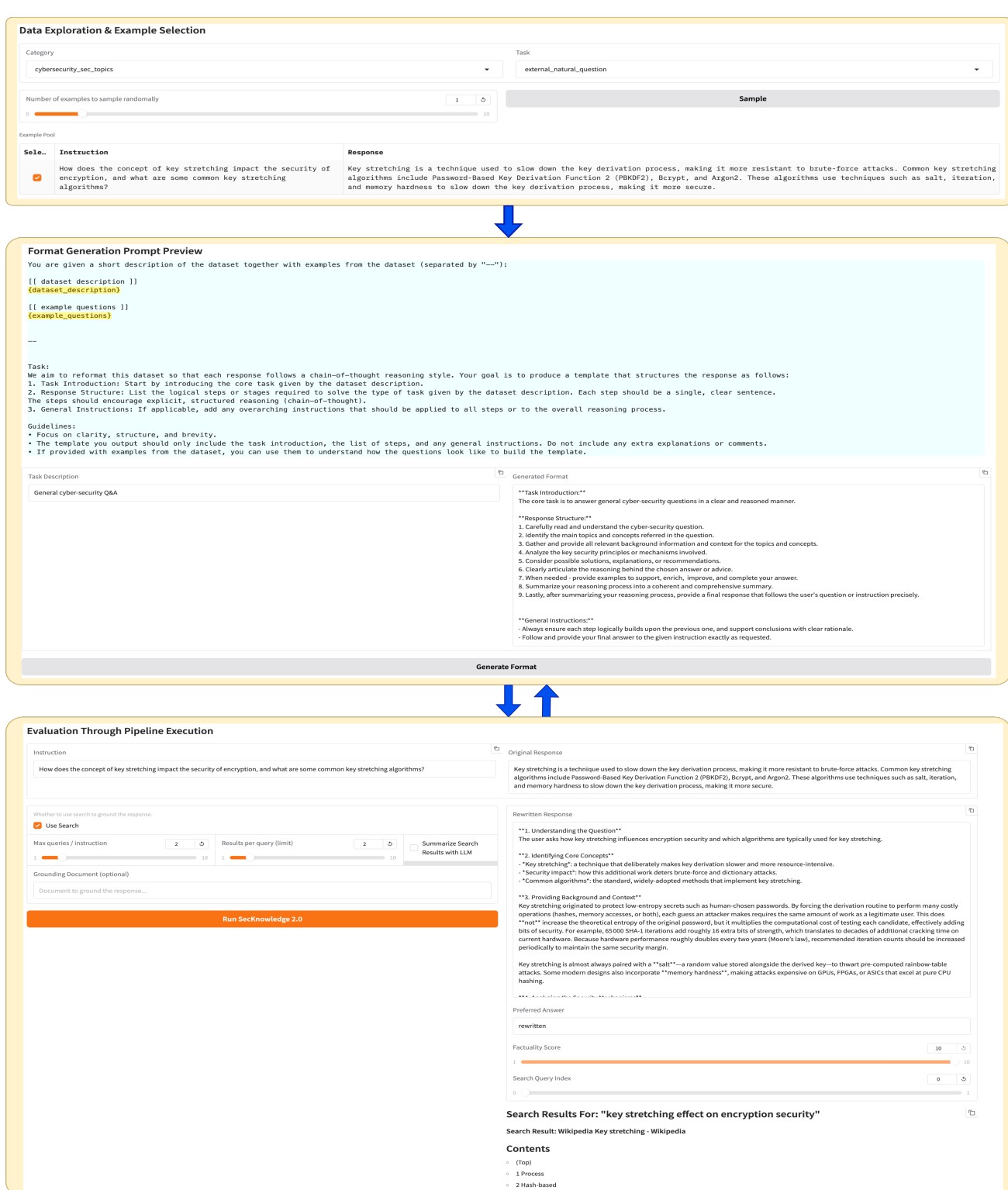

*Figure 6.* Screenshots from the UI designed for our format generation framework. First, the relevant task is selected, and random examples are sampled from the task data partition, then a candidate format is generated from them using a LLM and the prompt in blue, afterwards an expert can configure and run the pipeline, and edit the format if needed. The UI was developed using Gradio[6].

fine-tuning with *SecKnowledge 2.0* affects performance when starting from the base model versus a post-trained model. We

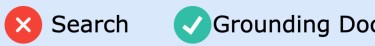

**Task**
**Name:** CAPECFlan@capec_mitigations
**Description:** List the mitigations of a specified attack pattern docummented in the MITRE CAPEC framework, if any.
**Format:**
The task is to identify and list mitigations for a specified attack pattern as documented in the MITRE CAPEC framework.

**Response Structure:**
1. Clearly identify the given attack pattern and briefly describe it.
2. Discuss the potential obstacles the attacker might bump into while attempting to execute this attack pattern.
3. Review the documentation given as grounding document to extract all mitigations associated with the attack pattern, if there are mitigations - describe how each mitigation targets a weak point in the attack. Otherwise - state that there are no known mitigations.
4. Summarize the mitigation strategies, and give general guidelines on how to mitigate this attack.

**General Instructions:**
- Make sure that you provide reasoning on how each mitigation adresses a weak point in the attack pattern.
- Rely on the CAPEC docummentation provided as grounding document.

❌ Search    ✅ Grounding Doc

*Figure 7.* An example task, specifically the one illustrated in Figure 3. A task consists of a name, a description, a format, whether it requires search, and whether it requires a grounding document.

observed an interesting phenomenon: directly fine-tuning the base model (i.e., Qwen3-8B-Base) yields significantly better results than relying on the post-trained model (Qwen3-8B) as the starting point. This effect is amplified on benchmarks that require additional reasoning to arrive at the final answer (e.g., CTIBench-RCM). On average, fine-tuning from the base model provides a 15.16% improvement across the key benchmarks, whereas starting from the post-trained model provides a 5.62% improvement relative to the Qwen3-8B baseline—corresponding to a $2.7\times$ larger gain when initializing from the base model. For open-ended benchmarks such as CTIBench-RCM, the difference is even more pronounced: the model fine-tuned from the base checkpoint achieves a 22.7% improvement, compared with 1.85% for the model fine-tuned from the post-trained checkpoint. Although limited in scope, this experiment empirically indicates that, given sufficiently high data quality, initializing from a base model enables more effective learning than starting from a post-trained checkpoint that has already undergone extensive supervised fine-tuning and alignment–yet further work is needed to systematically disentangle how data quality and data scale interact with the choice of starting checkpoint during fine-tuning.

*Table 4.* Comparing the improvement of fine-tuning our models when starting from base model vs. a post-trained model.

| Model | CTIBench MCQ | CTIBench RCM | SecEval | Avg. |
|---|---|---|---|---|
| Qwen 3 8b | 63.13 | 63.25 | 56.19 | 60.85 |
| CyberPal2.0-8B (trained from Qwen3-8B) | 68.90 | 65.10 | 65.42 | 66.47 |
| CyberPal2.0-8B (trained from Qwen3-8B-Base) | **75.15** | **85.95** | **66.93** | **76.01** |

**Incremental training methodology.** Lastly, consistent with the observations of Mitra et al. (2023); Levi et al. (2025), we empirically find that exposing the model to instructions of progressively increasing length—often correlated with task difficulty—enhances its learning capacity. Building on this principle, we adopt an incremental training methodology organized at the dataset level. Specifically, we first present the model with instructions from the original SecKnowledge dataset, followed by instructions from our new SecKnowledge 2.0 dataset.

**Additional training details.** We select the final checkpoint by validation loss on a held-out split extracted from the training set. We train for two epochs, as we observe diminishing returns and an increased risk of overfitting thereafter. Training was

conducted on a cluster of 12 NVIDIA A100 80 GB GPU nodes, and evaluation was performed on NVIDIA H100 80GB GPUs.

## C. Evaluation benchmarks, statistics, and analysis

### C.1. Evaluation benchmarks

**CTI-MCQ** (Alam et al., 2024) is a multiple choice question benchmark aimed at assessing LLMs' capabilities in understanding crucial cyber threat intelligence concepts including attack patterns, threat actors, APT campaigns, detection methods, mitigation strategies, common software vulnerabilities, attack pattern enumeration, alongside public CTI quizzes. This benchmark assess the breadth of CTI/domain knowledge; knowing frameworks/controls and when to apply them.

**CTI-RCM** CTI Root Cause Mapping (RCM) (Alam et al., 2024) identifies the underlying weakness(es) of a vulnerability by correlating CVE records and related bug tickets with CWE entries. Accurate root cause mapping is essential for guiding investments, policies, and practices aimed at addressing and eliminating these vulnerabilities. Strong LLM performance on CTI-RCM indicates grounded, document-linked reasoning and consistent, taxonomy-aware disambiguation—mapping real-world vulnerability evidence to the appropriate CWE(s) rather than relying on superficial keyword matches.

**SecEval** SecEval (Li et al., 2023) is a multiple-choice, multiple-option benchmark for evaluating LLMs' cybersecurity knowledge, with over 2,000 questions spanning nine domains. SecEval was constructed using OpenAI GPT-4 from authoritative sources (open-licensed textbooks, official platform security docs, OWASP guides, CWE, and MITRE ATT&CK/D3fend). This benchmark assesses the breadth and accuracy of security/domain knowledge and the ability to choose and apply the right frameworks, controls, detections, and mitigation to concrete scenarios.

**CyberMetric 2000** CyberMetric (Tihanyi et al., 2024) is a benchmark dataset for evaluating LLMs' knowledge in cybersecurity. The questions for the benchmark were created through a collaborative process, i.e., merging expert knowledge with LLMs. We used the 2000 questions dataset, verified by human evaluators, which covers a wide range of topics within cyber-security, validated by security experts. As questions come from standards-grounded material and were validated by certified practitioners (e.g., CISSP/CISM/OSCP), strong performance primarily evidences professional-level declarative cybersecurity knowledge—accurate recall of definitions, controls, and best practices across diverse subdomains, and the ability to reject plausible distractors under closed-book conditions.

**CISSP Exams** Introduced by (Levi et al., 2025), this benchmark uses exam-style questions from CISSP preparation materials to assess broad, professional cybersecurity knowledge across governance and risk, security architecture, operations, software and network security, and identity and access management. Items use plausible distractors and test principled reasoning and terminology rather than tool-specific tricks. A high score indicates strong declarative understanding, standards-aligned judgment, and the ability to separate best practices from common misconceptions under test conditions.

**Technical Weakness Impact Mapping** In CWE, each weakness, if successfully exploited, can lead to one or more technical impacts out of eight options: modify data, read data, DoS: unreliable execution, DoS: resource consumption, execute unauthorized code or commands, gain privileges / assume identity, bypass protection mechanism, and hide activities. This evaluation benchmark, introduced by (Levi et al., 2025), presents the model with CWEs and their descriptions, where the goal is to map each CWE to its related technical impact. A high score indicates taxonomy-aware understanding of how specific weakness patterns translate into concrete consequences, beyond surface keyword matching. Because a single CWE can map to multiple impacts and descriptions are often terse, the benchmark primarily measures consequence reasoning rather than exploit feasibility or business risk. It thus serves as an impact-from-weakness signal that complements severity or exploitability evaluations.

**Adversarial CTI** Levi et al. (2025) compiled an adversarial evaluation dataset from various MITRE ATT&CK sources to evaluate models on malicious software, campaigns, tactics, and corresponding detections and mitigations. Each input provides a question related to a specific MITRE instance, with the correct label being its corresponding source. To further challenge the models and test robustness, they introduced a novel adversarial attack for multiple-choice questions, where the attack chooses the false options that will confuse the model with the highest probability.

**CTI Detection and Mitigation** Introduced by Levi et al. (2025), this benchmark is designed to assess a model's ability to provide appropriate detections and mitigations for different attack tactics and techniques, attack patterns, weaknesses, and vulnerabilities.

**CTI Relationship Prediction** A major role of cyber threat management expert model is to comprehend the relationships between different CTI frameworks. This dataset (Levi et al., 2025) evaluates the ability to differentiate between false and correct relationships among CTI entities. For example, it presents the model with two entities (e.g., instances of CVE and CWE) and two possible explanations—one justifying why the entities are related and another explaining why they are not. The objective is for the model to reason and determine which explanation is correct.

## C.2. Evaluation statistics analysis

In this section, we provide details about the statistics and domain coverage of the security evaluation benchmarks. To evaluate the applicability of LLMs in cybersecurity, we first structured our evaluation set around domain-specific categories. The objective was to establish whether tasks align with areas such as threat intelligence, security operations, or identity and application security, and to provide a principled basis for mapping questions to specific domains of cybersecurity. A taxonomy-driven approach enables both standardized evaluation and benchmarking of model performance in a manner consistent with industrial and academic practices.

As part of this process, we explicitly built upon and extended two taxonomies: taxonomy of cybersecurity domains (Weerawardhena et al., 2025) and the SecEval benchmark dataset for security evaluation (Li et al., 2023). By synthesizing insights from both Weerawardhena et al. industrial perspective and SecEval's categorization, we constructed a unified taxonomy that captures enterprise security concerns while remaining aligned with established evaluation standards.

Our benchmarks are closely aligned with organizational security priorities, with a particular focus on *threat Intelligence*, *incident response*, *security operations*, *application security*, and *identity management*. This alignment ensures that evaluation outcomes are not only theoretically sound but also operationally relevant to real-world defensive strategies.

Through this integration, we ensured that our taxonomy is both conceptually rigorous and operationally validated, bridging the gap between industrial practice and academic research in cybersecurity evaluation.

### C.2.1. CYBERSECURITY CATEGORIES

We defined the following ten high-level categories, each with a set of sub-categories capturing specific security concerns:

1. **GCR (Governance, Risk, and Compliance)**
   - Risk Management & Security Strategy
   - Compliance and Regulations (e.g., GDPR, HIPAA)
   - Security Frameworks (e.g., NIST CSF, ISO 27001)
   - Security Policies & Architecture

2. **NetSec (Network, Infrastructure, and Endpoint Security)**
   - Perimeter and Network Security (Firewalls, VPNs, Wireless)
   - Endpoint Protection & MDM
   - IoT and OT/ICS Security
   - Mobile Security

3. **AppSec (Application and Software Security)**
   - Secure Software Development (DevSecOps)
   - Application & API Security
   - Vulnerability Management & Penetration Testing
   - Software Supply Chain Security (SBOM, third-party risk)

4. **CloudSec (Cloud and Data Security)**
   - Cloud Security Architecture & Tools
   - Identity and Access Management (IAM, PAM)
   - Data Loss Prevention & Privacy (DLP, encryption)
   - Cloud Compliance & Shared Responsibility Model

5. **IAM_ZT (Identity, Access, and Zero Trust)**

   - Authentication & Authorization (MFA, SSO, RBAC)
   - Identity Governance & Lifecycle
   - Zero Trust Architecture
   - Privileged Access Controls

6. **SecOps (Security Operations and Monitoring)**

   - SIEM, SOC, and Log Management
   - Security Automation & SOAR
   - Detection Engineering
   - Operational Resilience & Monitoring

7. **ThreatOps_IR (Threat Intelligence and Incident Response)**

   - Threat Detection, Analysis & Hunting
   - Threat Intelligence Platforms & IOCs
   - Advanced Persistent Threats (APTs)
   - Malware Techniques
   - Incident Response, Recovery & Digital Forensics

8. **CryptoSec (Cryptography and Secure Communications)**

   - Cryptographic Algorithms & PKI
   - Key Management
   - Post-Quantum Cryptography
   - Secure Protocols and Encryption Practices

9. **HumanSec (Security Awareness and Human Risk)**

   - Social Engineering Techniques (Phishing, Pretexting)
   - Insider Threat Management
   - Security Awareness Training
   - Behavioral Risk Analysis

10. **Other**

   - Cross-domain or emerging categories not covered above.

This taxonomy provided a structured basis for categorizing data and aligning evaluation with both research benchmarks and enterprise needs.

### C.2.2. MULTI-LABEL CLASSIFICATION

To carry out the mapping, we employed a large open-source language model (OSS-120B), which was deployed internally for reasons of data security and computational control. The model was prompted with a multi-label classification prompt, allowing it to assign tasks to one or more categories simultaneously. In Figure 9, we provide the prompt used to classify the evaluation benchmark to specific topics in cyber security. The results of our classification process detailed in Table 5 and is also visualized in Figure 4.

This choice was intentional: many real-world cybersecurity problems span across multiple domains (e.g., a phishing campaign may involve HumanSec, IAM_ZT, and ThreatOps_IR simultaneously). Restricting classification to single-label outputs would fail to capture these cross-cutting concerns.

*Table 5.* Counts by dataset and taxonomy category.

| dataset | GCR | NetSec | AppSec | CloudSec | IAM_ZT | SecOps | ThreatOps_IR | CryptoSec | HumanSec | Other |
|---|---|---|---|---|---|---|---|---|---|---|
| CTIBench-MCQ | 404 | 516 | 935 | 90 | 315 | 422 | 1409 | 112 | 64 | 11 |
| CTIBench-RCM | 87 | 28 | 1995 | 7 | 68 | 214 | 389 | 52 | 6 | 0 |
| SecEval | 706 | 620 | 1099 | 117 | 499 | 300 | 370 | 569 | 98 | 18 |
| CyberMetric-2000 | 722 | 563 | 250 | 37 | 253 | 182 | 304 | 373 | 172 | 119 |
| CISSP Exams | 102 | 42 | 16 | 0 | 26 | 20 | 30 | 20 | 18 | 22 |
| Weakness Impact Mapping | 62 | 11 | 332 | 0 | 61 | 3 | 26 | 29 | 4 | 6 |
| CTI Detect & Mitigate | 219 | 213 | 511 | 21 | 115 | 220 | 421 | 37 | 21 | 2 |
| Adv. CTI | 32 | 111 | 78 | 52 | 90 | 88 | 676 | 24 | 27 | 0 |
| CTI Relationship Prediction | 203 | 139 | 507 | 10 | 92 | 82 | 348 | 31 | 21 | 0 |
| **TOTAL** | **2537** | **2243** | **5723** | **334** | **1519** | **1531** | **3973** | **1247** | **431** | **178** |

### C.2.3. PROMPT VALIDATION USING SECEVAL CATEGORIES

To ensure the robustness and correctness of our classification prompt, we performed a validation against SecEval categories. Specifically, we tested whether the outputs of our multi-label classification aligned with SecEval's category definitions and coverage. This served as a quality assurance step for our classification pipeline. See Table 6 for agreement results between our classification pipeline and SecEval.

Through this process, we confirmed that the OSS-120B model, when guided by our taxonomy-driven prompt, consistently produced category assignments that were both internally coherent and externally validated against widely recognized benchmarks.

*Table 6.* Validation of our classification pipeline on SecEval categories and data sources, which were also classified using an OpenAI model (gpt-4o). We observe strong overall agreement with SecEval's classifications.

| Category Summary | Aligned % |
|---|---|
| ApplicationSecurity | 83.7 |
| Cryptography | 100.0 |
| MemorySafety | 99.9 |
| NetworkSecurity | 97.6 |
| PenTest | 87.1 |
| SoftwareSecurity | 97.1 |
| SystemSecurity | 88.4 |
| Vulnerability | 93.8 |
| WebSecurity | 84.4 |

## D. Evaluation templates and prompts

In Figure 10 we provide the prompt used for our evaluation process. Since we have both multi-choice as well as classification tasks, we replace the <EXPL>token with the specifics of each question type.

## E. Open source security models performance

We adopt the evaluation protocol from Section 4.3 and assess recent open-source cybersecurity models (Weerawardhena et al., 2025; Yu et al., 2025; SegoLily Labs; DeepHat-V1). We omit PRIMUS-Reasoning on CTIBench because its training set was distilled from CTIBench (Yu et al., 2025), making the comparison unfair. Since the baselines are 7B–8B, we report our 8B variant for a like-for-like comparison. Table 7 presents the full results: our 8B model outperforms all open-source baselines by a substantial margin.

*Table 7.* Evaluation results for CyberPal 2.0 8B compared to the recent opens-source cyber security models. (*) Primus-reasoning average is missing CTI benchmarks because its training set was distilled from CTIBench

| Model | CTI Bench MCQ | CTI Bench RCM | SecEval | Cyber Metric 2000 | CISSP Exams | Adv. CTI | Weakness Impact Mapping | CTI Detect & Mitigate | CTI Relationship Prediction | Avg. |
|---|---|---|---|---|---|---|---|---|---|---|
| DeepHat-v1-7B | 61.24 | 68.1 | 33.21 | 84.0 | 76.76 | 63.23 | 60.74 | 56.12 | 52.05 | 61.72 |
| Lily-Cybersecurity-7B-v0.2 | 55.31 | 42.9 | 37.14 | 80.0 | 68.18 | 58.45 | 52.43 | 39.71 | 46.34 | 53.38 |
| Primus-merged | 65.2 | 63.9 | 59.06 | 85.1 | 78.28 | 64.92 | 55.3 | 50.77 | 59.98 | 64.72 |
| Primus-reasoning | - | - | 53.03 | 86.05 | 73.23 | 64.78 | 53.58 | 52.24 | 58.79 | 63.01(*) |
| Foundation-Sec-8B-Instruct | 63.24 | 67.95 | 54.81 | 84.5 | 69.69 | 68.87 | 60.74 | 55.52 | 57.31 | 64.74 |
| **CyberPal-2.0-8B** | 75.15 | 85.95 | 66.93 | 89.85 | 88.89 | 87.61 | 71.06 | 70.26 | 87.66 | 80.37 |

# F. Comparison to Other Model Families

To rigorously validate that the performance of CyberPal 2.0 stems from our domain-specific alignment methodology rather than the inherent capabilities of the Qwen architecture, we conducted an ablation study against leading open-source models from diverse model families. As detailed in Table 8, we evaluated CyberPal-2.0-14B against Phi-4 (Abdin et al., 2024), Llama 4 Scout (Meta AI, 2025), Mixtral 8x22B (Mistral AI Team, 2024), Mistral Small 3.2 (Mistral AI Team, 2025), and DeepSeek-V3 (Liu et al., 2024).

Despite possessing significantly fewer parameters than competitors like DeepSeek V3 (685B) or Llama 4 Scout (109B), CyberPal 2.0 achieves the highest average performance across the suite (81.76%). It demonstrates particular dominance in complex reasoning tasks, such as CTI Relationship Prediction (92.93%) and CTI Bench RCM (86%), surpassing the closest general-purpose competitors by substantial margins. These results confirm that the SecKnowledge 2.0 training pipeline effectively generalizes high-level security reasoning capabilities that exceed the baselines of much larger models, regardless of their underlying architectural family.

*Table 8.* Evaluation results for CyberPal 2.0 14B compared to SOTA open source models from various families and architectures. It is evident that on average, our 14B model outperforms other model architectures despite being a fraction of their size. Models are sorted by number of parameters.

| Model | CTI Bench MCQ | CTI Bench RCM | SecEval | Cyber Metric 2000 | CISSP Exams | Adv. CTI | Weakness Impact Mapping | CTI Detect & Mitigate | CTI Relationship Prediction | Avg. |
|---|---|---|---|---|---|---|---|---|---|---|
| Microsoft Phi 4 | 68.22 | 64.00 | 63.73 | 91.00 | 83.33 | 66.76 | 68.48 | 64.03 | 60.28 | 69.98 |
| Mistral Small 3.2 24B Instruct 2506 | 68.82 | 68.05 | 67.47 | 91.60 | 87.37 | 74.37 | 65.90 | 67.19 | 76.22 | 74.11 |
| Llama 4 Scout 17B 16E Instruct | 69.46 | 71.95 | 67.84 | 92.50 | 87.37 | 79.01 | 66.48 | 68.58 | 75.32 | 75.39 |
| Mixtral 8x22B v0.1 | 62.81 | 66.70 | 65.65 | 87.75 | 82.32 | 71.41 | 59.89 | 61.66 | 77.51 | 70.63 |
| DeepSeek V3 | 73.35 | 72.45 | 63.68 | **93.65** | **91.41** | 78.73 | 69.63 | 68.97 | 65.17 | 75.23 |
| **CyberPal-2.0-14B** | **75.51** | **86.00** | **69.71** | 89.95 | 90.40 | **89.58** | **70.77** | **70.95** | **92.93** | **81.76** |

# G. Ablation Studies Additional Results

This section provides additional ablation results, in particular, this section measures the effects of the reformatting method, effect of back bone LLM and excluding components from the reformatting. We follow the same training recipe described in Section 4.1 for all models. Evaluation follows the protocol in Section 4.3: we use the prompt from Figure 10, extract final answers with regular expressions, and evaluate in a zero-shot setting with temperature set to zero.

### G.1. Effect of reformatting method

In Table 9, we report full results for the model trained on the original SecKnowledge dataset and for the model trained with the standard reformatted alignment method (Fan et al., 2024)

*Table 9.* Reformatting method ablation results

| Model | CTI Bench MCQ | CTI Bench RCM | SecEval | Cyber Metric 2000 | CISSP Exams | Adv. CTI | Weakness Impact Mapping | CTI Detect & Mitigate | CTI Relationship Prediction | Avg. |
|---|---|---|---|---|---|---|---|---|---|---|
| Qwen3-4B | 61.88 | 49.95 | 57.38 | 87.40 | 79.80 | 64.51 | 57.02 | 60.77 | 67.99 | 65.19 |
| SecKnowledge (Original) | 65.45 | 57.80 | 49.15 | **88.40** | **85.35** | **79.86** | 62.75 | 62.84 | 65.94 | 68.60 |
| Baseline Reformatting | 63.92 | 61.65 | 49.84 | 87.40 | 81.81 | 76.05 | 63.04 | **65.51** | **81.10** | 70.04 |
| CyberPal2.0-4B | **69.70** | **81.15** | **59.02** | 87.80 | 80.80 | 68.03 | **66.48** | 64.03 | 77.12 | **72.68** |

## G.2. Effect of Backbone LLM Replacement

Table 10, shows the results of switching the backbone LLM in our pipeline from gpt-oss-120b to Llama 4 maverick. All models were trained on Qwen3-4B-base.

*Table 10.* Results using CyberPal2.0-4B with different models as the reformatting component in our data generation pipeline. In the top row, we use Llama Maverick as the reformatting model, and in the bottom row, we use gpt-oss-120B.

| Model | CTI Bench MCQ | CTI Bench RCM | SecEval | Cyber Metric 2000 | CISSP Exams | Adv. CTI | Weakness Impact Mapping | CTI Detect & Mitigate | CTI Relationship Prediction | Avg. |
|---|---|---|---|---|---|---|---|---|---|---|
| CyberPal2.0-4B (Maverick reformatter) | **70.58** | 70.75 | 56.05 | **88.00** | **81.82** | **78.45** | **67.62** | 62.94 | 74.68 | 72.32 |
| CyberPal2.0-4B (gpt-oss-120 reformatter) | 69.70 | **81.15** | **59.02** | 87.80 | 80.80 | 68.03 | 66.48 | **64.03** | **77.12** | **72.68** |

## G.3. Effect of search component in the pipeline

One of the key components of our pipeline is the search module, which ensures that model outputs remain accurate and reliable. Table 11 presents the results for models trained without the search component compared to those trained with it. Removing the search component leads to an average performance drop of approximately 3%, confirming that retrieval consistently enhances overall accuracy.

*Table 11.* The effect of removing the search component from the reformatting pipeline

| Model name | Pipeline | CTI-MCQ | CTI-RCM | Avg. |
|---|---|---|---|---|
| CyberPal 2.0 4B | No search | 67.82 | 80.26 | 71.11 |
| | Full pipeline | 69.70 | 81.15 | 72.68 |
| CyberPal 2.0 8B | No search | 72.75 | 84.90 | 76.87 |
| | Full pipeline | 75.15 | 85.95 | 80.37 |
| CyberPal 2.0 14B | No search | 74.99 | 85.75 | 78.83 |
| | Full pipeline | 75.51 | 86.00 | 81.76 |
| CyberPal 2.0 20B | No search | 74.99 | 87.1 | 78.56 |
| | Full pipeline | 75.71 | 84.7 | 80.33 |

## H. LLMaaJ Experiment details and additional results

To assess answer quality, we used *LLM-as-a-Judge* (LLMaaJ) (Zheng et al., 2023). Thirty cybersecurity experts authored 115 open-ended questions: spanning command-line risk assessment, enterprise security, general cybersecurity, network security, and CTI-related topics. Specifically, the security experts constructed 20 questions related cyber threat intelligence, 20 questions related to security vulnerabilities, 20 questions related to network security, 16 general security questions, 20 questions related to enterprise security, and 19 questions related to command line risk assessment.

**Pairwise comparison with grounding —** The judge receives a question, two answers, and a carefully collected grounding documents that contains all relevant information to answer the question. The judge should decide which answer is better. The prompt provided to the LLMaaJ is provided in Figure 11 .

**Evaluation process —** We use OpenAI's o3 (OpenAI, 2025) as the LLM-as-a-judge. The judge evaluates each answer pair along six dimensions —*Contextual Accuracy* (highest priority), *Helpfulness*, *Relevance*, *Conciseness*, *Completeness*, and *length bias* (Gu et al., 2024) then issues a verdict: A better than B, B better than A, tie, or both bad. To mitigate positional bias in LLM-as-a-judge settings (Wang et al., 2023; Zheng et al., 2023), we run the comparison twice with the answers swapped. For each permutation, a model receives a score of 3 if its answer is preferred by the judge, 1 for tie, and 0 for loss; if the preferences flip across orders, the pair will effectively contribute 0 as $3 - 3 = 0$. We also record ties and losses separately, though these were rare in our experiments.

**Alignment with human preferences —** To validate the judge, we measured agreement with Thirty cybersecurity human experts and found that, with proper grounding, o3 aligns with human preferences in over 90% of cases. Without proper grounding, alignment decreases to 80%.

In Figure 5, we report our LLMaaJ results across all questions. Additionally, in Figures 12, 13, and 14 we report LLMaaJ results per category. As can be observed, our model is preferable by a large margin across all the tested categories.

## I. Model Quantization

As a deployment-oriented baseline, we also evaluated quantization using bitsandbytes (Dettmers et al., 2022) by loading models directly in 8-bit and 4-bit modes, without any calibration or advanced schemes which are shown to perform better than out-of-the-box quantization (Frantar et al., 2022). Across our evaluation suite, 8-bit loading resulted in a negligible drop of 0.36% for the 4B model and 0.84% for the 8B model — relative to full precision. Moving to 4-bit, both models saw a larger drop around 4% absolute for the 8B model and 2.78% for the 4B model. Importantly, both quantized modes remained clearly superior to the instruction-tuned baseline, which was not trained using the SecKnowledge 2.0 pipeline. These results suggest that the benefits of fine-tuning largely persist under straightforward low-precision inference, with 8-bit serving as a particularly safe, low-overhead option for memory-constrained deployment and 4-bit serves as a good choice for fast inference or low resource settings, while still keeping high cyber security knowledge.

*Table 12.* Quantization results. Cells in quantized rows show $\Delta$ value with (arrow $\downarrow$ OR $\uparrow$) , where $\Delta=|\text{Full} - \text{Quantized}|$.

| Model | CTI MCQ | CTI RCM | SecEval | Cyber Metric | CISSP | Adv. CTI | Weakness Impact Mapping | CTI Detect & Mitigate | CTI Relationship Prediction | Avg. |
|---|---|---|---|---|---|---|---|---|---|---|
| **4B models** | | | | | | | | | | |
| Qwen 3-4B | 61.88 | 49.95 | 57.38 | 87.40 | 79.80 | 64.51 | 57.02 | 60.77 | 67.99 | 65.19 |
| CyberPal 2.0 | 69.70 | 81.15 | 59.02 | 87.80 | 80.80 | 68.03 | 66.48 | 64.03 | 77.12 | 72.68 |
| CyberPal 2.0 (8-bit) | (↓0.39) | (↑0.60) | (↓1.00) | (↓0.75) | (↓1.51) | (↑0.15) | (↓0.28) | (↓1.18) | (↑1.15) | (↓0.36) |
| CyberPal 2.0 (4-bit) | (↓3.88) | (↓4.15) | (↓4.11) | (↓4.6) | (↓3.54) | (↑0.85) | (↓3.73) | (↓3.16) | (↑1.28) | (↓2.78) |
| **8B models** | | | | | | | | | | |
| Qwen 3-8B | 63.13 | 63.25 | 56.19 | 88.45 | 83.33 | 64.93 | 53.58 | 59.88 | 60.67 | 65.93 |
| CyberPal 2.0 | 75.15 | 85.95 | 66.93 | 89.85 | 88.89 | 87.61 | 71.06 | 70.26 | 87.66 | 80.37 |
| CyberPal 2.0 (8-bit) | (↓1.08) | (↓1.10) | (↑1.78) | (0.00) | (↓3.03) | (↓2.11) | (↑0.57) | (↓1.68) | (↓0.9) | (↓0.84) |
| CyberPal 2.0 (4-bit) | (↓4.65) | (↓3.10) | (↓2.97) | (↓3.35) | (↓3.54) | (↓9.44) | (↑2.01) | (↓4.45) | (↓3.86) | (↓4.15) |

# J. Generalization to Real world Use Cases

## J.1. Threat Reports to TTP Mapping

To assess the model's ability to generalize to sources outside our training distribution, we built a benchmark using independently collected external threat-analysis reports, including technical write-ups, industry blogs, and vendor whitepapers. These documents were drawn from public sources not used in our training pipeline, allowing us to evaluate how well the model handles unseen threat reports.

Each report is paired with its corresponding attack technique through the existing mapping provided on the report's associated campaign entry in public threat-intelligence repositories. Using this information, we formulate a multiple-choice classification task: the model receives the raw, unstructured report text and must select the correct technique from a set of candidate options. The model did not encounter this type of report-classification task during training, making it a novel reasoning setting in addition to the reports themselves being unseen.

We report CyberPal 2.0 performance on this benchmark in Table 13.

*Table 13.* Threat reports to TTPs: CyberPal 2.0 models compared to their corresponding baselines.

| Model Name | Score |
|---|---|
| Qwen3-4B | 74.00 |
| **CyberPal-2.0-4B** | **82.75** (+8.75) |
| Qwen 3 8b | 74.00 |
| **CyberPal-2.0-8B** | **78.25** (+4.25) |
| Qwen 3 14b | 79.25 |
| **CyberPal-2.0-14B** | **85.25** (+6.00) |
| gpt-oss-20B | 76.75 |
| **CyberPal-2.0-20B** | **79.50** (+2.75) |

## J.2. CyberSOCEval benchmark

We further evaluate our models on *CyberSOCEval* (Deason et al., 2025), a recently released suite within CyberSecEval 4 that targets core SOC workflows. The benchmark comprises two multiple-choice tasks **Malware Analysis** and **Threat Intelligence Reasoning** which scores models by exact-match accuracy (all and only the correct options) And Jaccard score (intersection of correct answers (the size of the intersection between the predicted and gold answer sets divided by the size of their union).

- **Malware Analysis** — Questions are grounded in real Sandbox detonation reports (e.g., process trees, extracted files, network activity). The task probes an LLM's ability to interpret low-level telemetry and identify malicious behavior (Deason et al., 2025).

- **Threat Intelligence Reasoning** — Questions are derived from full-page threat-intel reports (provided as page images), assessing an LLM's capacity to extract actionable insights (e.g., adversary tactics, MITRE ATT&CK mappings, targeted sectors) beyond surface-level comprehension (Deason et al., 2025).

Since malware analysis tasks require extremely long context windows (up to 128k tokens for full prompts and approximately 32k for truncated ones), and our models were trained with a maximum sequence length of 8k tokens, we chose not to report results on this benchmark. The full benchmark results are presented in Table 14. To ensure consistent evaluation conditions, we re-ran all experiments using both LLaMA 4 and GPT-4o under identical settings, in particularly identical prompts which allows for a fair, apples-to-apples comparison across models. Our results demonstrate that our model consistently outperforms the strongest baselines by a substantial margin, while remaining competitive with significantly larger open models, some up to ten times our model's size narrowing the performance gap to roughly 4% on average.

*Table 14.* CyberSocEval Threat Intelligence reasoning task

| Model | Accuracy | Jaccard |
|---|---|---|
| Qwen3-4B | 5.95 | 10.12 |
| CyberPal-2.0-4B | 19.39 | 51.85 |
| Qwen3-8B | 42.86 | 58.59 |
| CyberPal-2.0-8B | 38.61 | 65.60 |
| Qwen3-14B | 43.54 | 63.12 |
| CyberPal-2.0-14B | 45.07 | 67.78 |
| LLaMa-4-Maverick | 54.25 | 72.57 |
| LLaMa-4-Scout | 50.34 | 69.90 |
| GPT-4o | 53.57 | 73.19 |

## J.3. CVE Reassessment benchmark

One real-world use case we are currently dealing with, both internally and with our clients, is how to reassess a CVE's applicability and severity score for a specific package, deployment configuration, service, etc.

Each CVE is associated with a Common Vulnerability Scoring System (CVSS) base score, which provides additional guidance about the vulnerability by scoring constant aspects such as: Attack Vector, Attack Complexity, User Interaction, Privileges Required, Scope, Confidentiality, Integrity, and Availability.

A single CVE can affect many services and packages, and the CVSS score, derived from CVSS, often reflects a broad perspective and worst-case scenarios. In practice, CVSS base scores may vary for each vendor's version, depending on the version they ship, how they ship it, the platform, and even how the software is compiled. This makes it difficult for third-party vulnerability databases (such as NVD), which can assign only a single CVSS base score per vulnerability, and also complicates comparison with vendors who score based on how the vulnerability manifests in their own products.

Therefore, a key real-world use case is to reassess the CVSS score of a given CVE under specific constraints. For example, a NIST CVE may receive a High Impact score in general, but for a specific service or product that runs with low privileges, the effective Impact score may be lower.

To study this, we constructed a benchmark of real CVEs (from 2025, to avoid data contamination) along with their original CVSS vectors. For each CVE, we paired specific products and services affected by that CVE and obtained new CVSS scores defined by security experts specifically for those products. The goal of this benchmark is to test how well LLMs can reassess the CVSS vector for concrete packages, services, and deployment configurations.

Results are reported as MAD (Mean Absolute Deviation), the same metric used in CTI-Bench, normalized to the 0–1 range (1-[SCORE/10]). We also punish for bad responses which do not contain the final CVSS score by given the highest score of 10. As can be seen from tables 15 and 16, our models show impressive improvements compared to both the baselines and other open-source cybersecurity LLMs.

*Table 15.* CVE Reassessment results of **open-source cybersecurity LLMs vs. CyberPal-2.0-8b.**

| Model Name | MAD (normalized) |
|---|---|
| Lily-Cyber-7B-v0.2 | 0.635 |
| Llama-Primus-merged | 0.802 |
| Llama-Primus-reasoning | 0.807 |
| DeepHat-V1-7B | 0.823 |
| Foundation-Sec-8B-Instruct | 0.698 |
| **CyberPal-2.0-8B** | **0.834** |

*Table 16.* CVE Reassessment results of **CyberPal-2.0-8b vs. Baselines.**

| Model Name | MAD (normalized) |
|---|---|
| Qwen3-4b | 0.783 |
| CyberPal-2.0-4b | 0.830 |
| Qwen3-8b | 0.825 |
| CyberPal-2.0-8b | 0.834 |
| Qwen3-14 | 0.740 |
| CyberPal-2.0-14b | 0.833 |
| gpt-oss-20b | 0.738 |
| CyberPal-2.0-20b | 0.834 |

## J.4. Secure Code Generation Benchmarks

When developing a cybersecurity-oriented language model, an important risk emerges: models trained to reason about security may inadvertently generate insecure or vulnerability-prone code. This concern is especially relevant for LLMs designed to assist security practitioners, where users may rely on generated code for analysis, testing, or defensive automation. To ensure that our CyberPal 2.0 models do not introduce insecure coding patterns, we systematically evaluate them using established secure code generation benchmarks.

**CYBERSECEVAL** (Bhatt et al., 2023) introduces two complementary evaluation paradigms designed to measure how LLMs reproduce or generate insecure coding patterns in realistic development settings. Both benchmarks are built on a shared methodology: real-world insecure code is automatically identified in open-source repositories using the Insecure Code Detector (ICD), a static analysis framework containing 189 rules across 50 CWE categories, and these vulnerabilities are transformed into prompts that probe model behavior. By applying the same detection pipeline to model outputs, the benchmarks jointly characterize the security reliability of LLMs under different prompting modalities.

- **Insecure Code Generation — Autocomplete** Evaluates whether a model continues code with insecure patterns when given preceding lines taken from insecure open-source snippets, reflecting risks in code-completion workflows.

- **Insecure Code Generation — Instruct** Evaluates whether a model produces insecure code when responding to natural-language instructions generated from insecure code segments, reflecting risks in instruction-based coding workflows.

*Table 17.* Secure Code Generation - Autocomplete, Instruct, and Average Pass Rates. Pass rate measures the percentage of test cases in which a model avoids reproducing insecure coding practices, as defined by the ICD.

| Model Name | Autocomplete | Instruct | Average |
|---|---|---|---|
| Qwen-3-4B | 78.81 | 80.13 | 79.47 |
| CyberPal 2.0 Qwen-3-4B | 76.10 | 67.16 | 71.63 |
| Qwen-3-8B | 79.38 | 79.77 | 79.58 |
| CyberPal 2.0 Qwen-3-8B | 72.81 | 69.07 | 70.94 |
| Qwen-3-14B | 77.45 | 74.54 | 75.99 |
| CyberPal 2.0 Qwen-3-14B | 73.33 | 66.33 | 69.83 |
| gpt-oss-20b | 70.25 | 68.53 | 69.39 |
| CyberPal 2.0 gpt-oss-20b | 67.90 | 63.95 | 65.93 |

Across all evaluated scales, CyberPal 2.0 demonstrates solid secure code generation performance, with results that remain close to those of the instruction-tuned comparison models - even though those models were further trained to provide safer responses, including in the context of secure code generation, while CyberPal 2.0 is trained directly from base models. On average, CyberPal 2.0 shows only a 6.52% reduction relative to the aligned models, a modest change that aligns with

expectations when adapting a model toward specialized cybersecurity reasoning. Despite this shift, CyberPal 2.0 retains most of the secure-coding characteristics of its reference models, indicating that the specialization process preserves core code-safety behavior while enabling substantially enhanced cybersecurity capabilities.

## K. Training time analysis

To provide a clearer view of computational efficiency, Table 18 reports a partial overview of the training durations for our models. All models were trained on NVIDIA A100 GPUs (80 GB) using a context length of 8192 tokens, a gradient accumulation step of 32, and an effective batch size of 3. We employed a boundary-preserving grouping and sequence-packing strategy to maximize hardware utilization and minimize idle time during training.

*Table 18.* Training time analysis

| Model size | GPU Count | Training hours |
|---|---|---|
| 4B | 40 | 1 Day, 6 hours and 40 minutes |
| 8B | 40 | 2 Day, 19 hours and 18 minutes |

## L. Inference latency

We benchmarked all model variants across quantization levels (FP16, 8-bit, 4-bit) and batch sizes (1, 4, 8) using a representative cyber security-style prompt (512 input tokens, 128 generated tokens) on a single NVIDIA H100-80GB GPU. For each configuration, we measured end-to-end generation latency, approximate time-to-first-token (TTFT), throughput (tokens per second), and peak GPU memory footprint (including weights and KV-cache). Results are reported in seconds to highlight practical latency ranges. The measurements show that FP16 inference, as expected has the highest token per sec rate, while 8-bit quantization reduces memory usage by nearly 40% with moderate throughput trade-offs. Batch scaling demonstrates nearly linear throughput gains until GPU saturation, indicating stable memory behavior across configurations. These findings confirm that quantized models can meet real-time operational requirements (e.g., less then 100 ms per token generation) while significantly reducing hardware cost.

Table 19 summarizes the observed performance. Our 8-bit variants consistently exhibited higher end-to-end latency compared to both 4-bit and FP16 configurations. While their overall throughput (tokens per second) remained competitive, the TTFT was substantially longer—often by an order of magnitude. Although we cannot definitively isolate the cause, prior work suggests that some hardware backends perform runtime de-quantization of 8-bit weights (Zhang et al., 2024), introducing additional computational overhead. Moreover, a recent large-scale empirical study reported that "quantization does not always reduce latency in online serving." (Shi & Ding, 2025). Together, these observations explain why our 8-bit inference runs showed higher latency than FP16 despite achieving a smaller memory footprint.

## M. Use of Large Language Models (LLMs)

When writing the paper, we used LLMs to help us find grammar errors and polish sentences that needed further clarifications. No further usage was done using LLMs while writing the paper.

## N. Qualitative results

We include additional examples of the training examples before and after pipeline in Figure 15 and Figure 16

*Table 19.* Inference latency

| Model Label | Bit Mode | Batch Size | Prompt Length | Generation Length | Latency (Sec) | TTFT (Sec) | Tokens Per Sec | Memory peak (GB) |
|---|---|---|---|---|---|---|---|---|
| CyberPal-4B | 4bit | 1 | 512 | 128 | 5.34 | 0.042 | 23.95 | 3.27 |
| CyberPal-4B | 4bit | 4 | 512 | 128 | 7.83 | 0.060 | 65.39 | 3.37 |
| CyberPal-4B | 4bit | 8 | 512 | 128 | 7.64 | 0.06 | 134.07 | 3.46 |
| CyberPal-4B | 8bit | 1 | 512 | 128 | 22.45 | 0.175 | 5.70 | 4.83 |
| CyberPal-4B | 8bit | 4 | 512 | 128 | 22.83 | 0.178 | 22.43 | 4.89 |
| CyberPal-4B | 8bit | 8 | 512 | 128 | 18.60 | 0.145 | 55.05 | 4.99 |
| CyberPal-4B | fp16 | 1 | 512 | 128 | 4.09 | 0.032 | 31.33 | 8.440 |
| CyberPal-4B | fp16 | 4 | 512 | 128 | 9.68 | 0.076 | 52.91 | 8.52 |
| CyberPal-4B | fp16 | 8 | 512 | 128 | 4.86 | 0.038 | 210.52 | 8.62 |
| CyberPal-8B | 4bit | 1 | 512 | 128 | 5.30 | 0.041 | 24.15 | 6.53 |
| CyberPal-8B | 4bit | 4 | 512 | 128 | 7.56 | 0.059 | 67.72 | 6.62 |
| CyberPal-8B | 4bit | 8 | 512 | 128 | 7.55 | 0.059 | 135.70 | 6.72 |
| CyberPal-8B | 8bit | 1 | 512 | 71 | 8.23 | 0.116 | 8.63 | 9.55 |
| CyberPal-8B | 8bit | 4 | 512 | 128 | 16.42 | 0.128 | 31.18 | 9.58 |
| CyberPal-8B | 8bit | 8 | 512 | 128 | 16.97 | 0.133 | 60.33 | 9.66 |
| CyberPal-8B | fp16 | 1 | 512 | 59 | 1.87 | 0.032 | 31.53 | 18.83 |
| CyberPal-8B | fp16 | 4 | 512 | 59 | 1.90 | 0.032 | 124.15 | 18.88 |
| CyberPal-8B | fp16 | 8 | 512 | 59 | 1.89 | 0.032 | 249.78 | 18.95 |
| CyberPal-14B | 4bit | 1 | 512 | 21 | 0.99 | 0.047 | 21.17 | 14.69 |
| CyberPal-14B | 4bit | 4 | 512 | 21 | 1.99 | 0.095 | 42.23 | 14.72 |
| CyberPal-14B | 4bit | 8 | 512 | 21 | 1.87 | 0.089 | 89.90 | 14.76 |
| CyberPal-14B | 8bit | 1 | 512 | 128 | 19.15 | 0.150 | 6.68 | 20.48 |
| CyberPal-14B | 8bit | 4 | 512 | 128 | 19.34 | 0.151 | 26.48 | 20.51 |
| CyberPal-14B | 8bit | 8 | 512 | 128 | 18.62 | 0.145 | 55.01 | 20.55 |
| CyberPal-14B | fp16 | 1 | 512 | 128 | 5.790 | 0.045 | 22.11 | 33.50 |
| CyberPal-14B | fp16 | 4 | 512 | 128 | 7.36 | 0.057 | 69.59 | 33.58 |
| CyberPal-14B | fp16 | 8 | 512 | 128 | 5.26 | 0.041 | 194.78 | 33.69 |
| CyberPal-20B | 4bit | 1 | 512 | 128 | 5.06 | 0.04 | 25.31 | 40.98 |
| CyberPal-20B | 4bit | 4 | 512 | 128 | 5.62 | 0.044 | 91.14 | 41.15 |
| CyberPal-20B | 4bit | 8 | 512 | 128 | 5.55 | 0.043 | 184.39 | 41.37 |
| CyberPal-20B | 8bit | 1 | 512 | 128 | 10.96 | 0.086 | 11.68 | 41.26 |
| CyberPal-20B | 8bit | 4 | 512 | 128 | 11.90 | 0.093 | 43.03 | 41.43 |
| CyberPal-20B | 8bit | 8 | 512 | 128 | 13.34 | 0.104 | 76.79 | 41.65 |
| CyberPal-20B | fp16 | 1 | 512 | 128 | 4.54 | 0.035 | 28.18 | 49.95 |
| CyberPal-20B | fp16 | 4 | 512 | 128 | 5.56 | 0.043 | 92.01 | 50.11 |
| CyberPal-20B | fp16 | 8 | 512 | 128 | 5.890 | 0.046 | 173.9 | 50.34 |

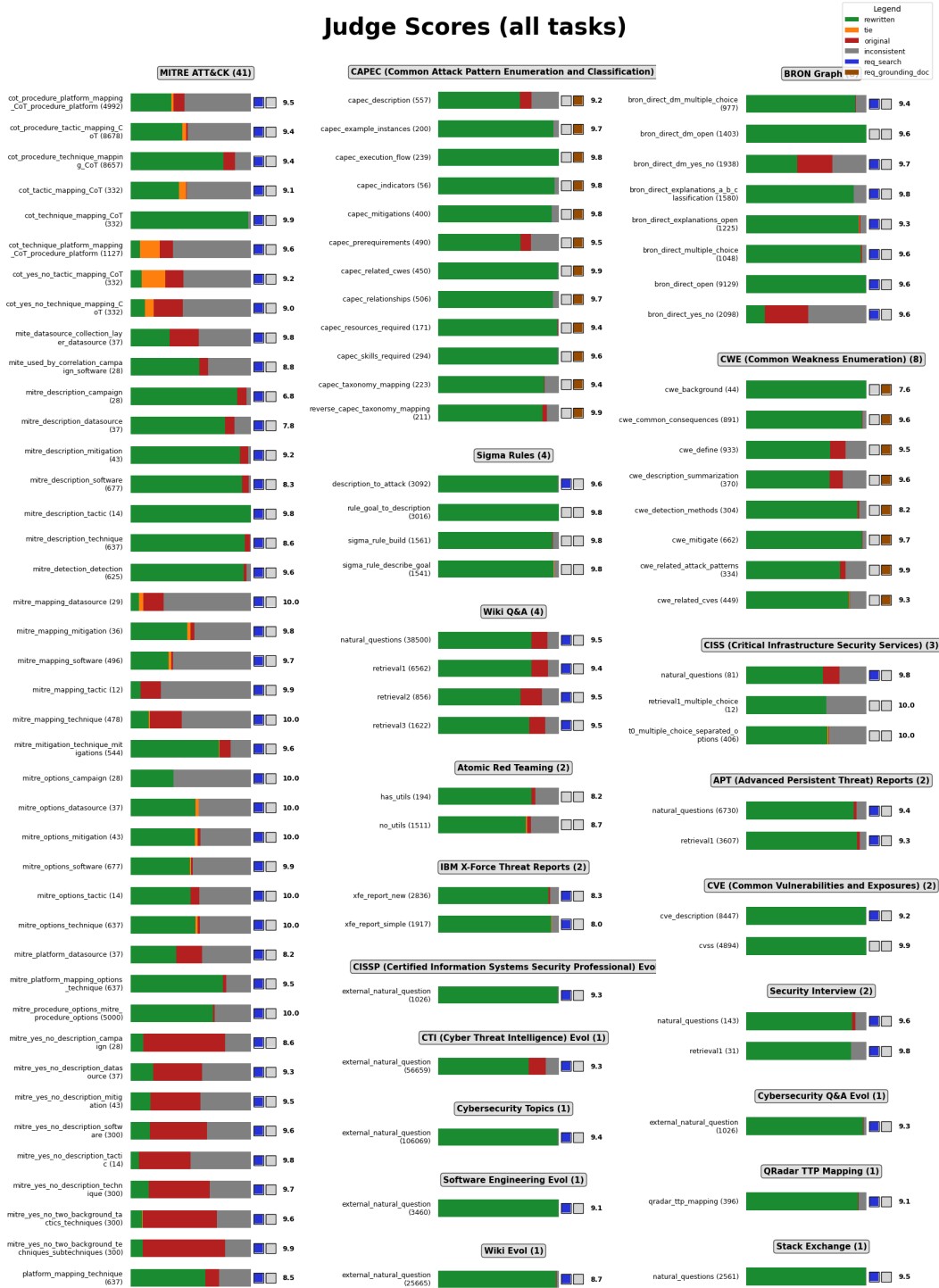

*Figure 8.* Data generation quality assessment scores for *SecKnowledge 2.0*, broken down by task. Each bar is color-coded to indicate **readability** outcomes (green for rewritten, red for original, orange for tie, and gray for position inconsistency). The boxes to the right of each bar shows the context requirements (blue box for a task that requires web search, brown box for a task that requires a grounding document), and the number to the right of those boxes denotes the average **factuality** score. The framed labels above groups of tasks indicates their parent category (categories ending with "Evol" contain examples synthetically generated by the second phase of *SecKnowledge*). The number in parentheses after each category indicates how many tasks belong to that category, while the number in parentheses after each task represents the count of instructions within that task.

You are a cybersecurity expert with deep knowledge across all domains of cybersecurity.
Your task is to assign each cybersecurity evaluation benchmark question to one or more predefined categories, using only the question text and its suggested answer options.

**Important:**
A question may belong to more than one category.
You must provide a brief explanation for each category decision.
Output must strictly follow the JSON format below.

**Cybersecurity Categories:**

Category 1: GCR
sub_categories:
- Risk Management & Security Strategy
- Compliance and Regulations (e.g., GDPR, HIPAA)
- Security Frameworks (e.g., NIST CSF, ISO 27001)
- Security Policies & Architecture
Category 2: NetSec
sub_categories:
- Perimeter and Network Security (Firewalls, VPNs, Wireless)
- Endpoint Protection & MDM
- IoT and OT/ICS Security
- Mobile Security
Category 3: AppSec
sub_categories:
- Secure Software Development (DevSecOps)
- Application & API Security
- Vulnerability Management & Penetration Testing
- Software Supply Chain Security (SBOM, third-party risk)
Category 4: CloudSec
sub_categories:
- Cloud Security Architecture & Tools
- Identity and Access Management (IAM, PAM)
- Data Loss Prevention & Privacy (DLP, encryption)
- Cloud Compliance & Shared Responsibility Model
Category 5: IAM_ZT
sub_categories:
- Authentication & Authorization (MFA, SSO, RBAC)
- Identity Governance & Lifecycle
- Zero Trust Architecture
- Privileged Access Controls
Category 6: SecOps
sub_categories:
- SIEM, SOC, and Log Management
- Security Automation & SOAR
- Detection Engineering
- Operational Resilience & Monitoring
Category 7: ThreatOps_IR
sub_categories:
- Threat Detection, Analysis & Hunting
- Threat Intelligence Platforms & IOCs
- Advanced Persistent Threats (APTs)
- Malware Techniques
- Incident Response, Recovery & Digital Forensics
Category 8: CryptoSec
sub_categories:
- Cryptographic Algorithms & PKI
- Key Management
- Post-Quantum Cryptography
- Secure Protocols and Encryption Practices
Category 9: HumanSec
sub_categories:
- Social Engineering Techniques (Phishing, Pretexting)
- Insider Threat Management
- Security Awareness Training
- Behavioral Risk Analysis
Category 10: Other

**Input format:**

Here is the question:
Choose the correct option for the following question: Which of the following mitigations involves preventing applications from running that haven't been downloaded from legitimate repositories?
A: Audit
B: Execution Prevention
C: Operating System Configuration
D: User Account Control

Here is the answer:
B

**Output format:**
{
    explanation_GCR: "reasoning here", "GCR": "yes/no",
    explanation_NetSec: "reasoning here", "NetSec": "yes/no",
    explanation_AppSec: "reasoning here", "AppSec": "yes/no",
    explanation_CloudSec: "reasoning here", "CloudSec": "yes/no",
    explanation_IAM_ZT: "reasoning here", "IAM_ZT": "yes/no",
    explanation_SecOps: "reasoning here", "SecOps": "yes/no",
    explanation_ThreatOps_IR: "reasoning here", "ThreatOps_IR": "yes/no",
    explanation_CryptoSec: "reasoning here", "CryptoSec": "yes/no",
    explanation_HumanSec: "reasoning here", "HumanSec": "yes/no",
    explanation_FutureSec: "reasoning here", "FutureSec": "yes/no",
    explanation_Other: "reasoning here", "Other": "yes/no",
}

*Figure 9.* The prompt used to classify the examples in SecKnowledge 2.0 into cybersecurity topics

**You must produce TWO sections for you answer in this exact order:**
1) Explanation — a logical step by step rationale for choosing the correct option.
2) '#### Final Answer: {TOKEN}' — a single line with the final answer, <EXPL>.

**Formatting rules:**
- Start with the line: 'Explanation:'.
- Write the explanation for selecting the correct option.
- The last line MUST be exactly: '#### Final Answer: {TOKEN}'.

**Output template (follow literally):**
Explanation:
<your explanation>
#### Final Answer: {TOKEN}

*Figure 10.* The prompt used to guide the LLMs during the evaluation process. <EXPL> refers to the specific task type (multi-choice, classification, etc.) and is replaced at runtime with explanation about the format of the specific question.

Act as an impartial and meticulous evaluator tasked with assessing the quality of responses provided by two AI assistants to the user prompt outlined below.
You will receive responses from Assistant A and Assistant B. Your responsibility is to determine which assistant's response is superior by conducting a thorough analysis based on the provided criteria.

## Evaluation Process:
1. **Contextual Accuracy (Highest Priority):**
   - Verify the correctness of the responses with respect to the user Question and provided Context.
   - Identify any inaccuracies, errors, or misinterpretations. Responses with factual inaccuracies or contradictions to the given Context are heavily penalized.

2. **Helpfulness:**
   - Assess whether the response appropriately addresses the user's prompt or instructions.

3. **Relevance:**
   - Determine if all parts of the response align closely with the user's question or request.
   - Penalize extraneous or off-topic information.

4. **Conciseness:**
   - Evaluate the clarity and brevity of the response.
   - Ensure the assistant avoids unnecessary verbosity while maintaining the completeness of the answer.

5. **Completeness:**
   - Identify if any important information is missing in the assistants' answers that would be beneficial to include when responding to the user prompt.

6. **Avoid answers' length bias:**
   - Do not favor a specific answer simply because it is longer. Choose the better answer based on correctness.

**Remember:**
* **Primary Criterion:** The most important factor is how accurately each answer reflects the content, reasoning, and details of the provided Context. The answer closest to the Context—both in terms of factual correctness and conceptual alignment—should be preferred.
* **Secondary Criteria:** While Helpfulness, Relevance, and Completeness are important, they should only influence your decision when both answers are equally close to the Context.
* **Do not prefer longer answers:** You should not prefer a specific answer simply since it is longer. Judge the answers based on the previous steps, but do not be biased to longer answers.

## Final Verdict:
After analyzing the responses based on the criteria above, provide a concise explanation supporting your judgment. Conclude with one of the following verdicts:

1. **Assistant A is better: [[A>B]]**
2. **Assistant B is better: [[B>A]]**
3. **Tie: [[A=B]]**
4. **Both assistants failed to answer correctly: [[B<>A]]**

Example Final Verdict: "My final verdict is a Tie: [[A=B]]"

*Figure 11.* LLM-as-Judge prompt used for pairwise comparison.

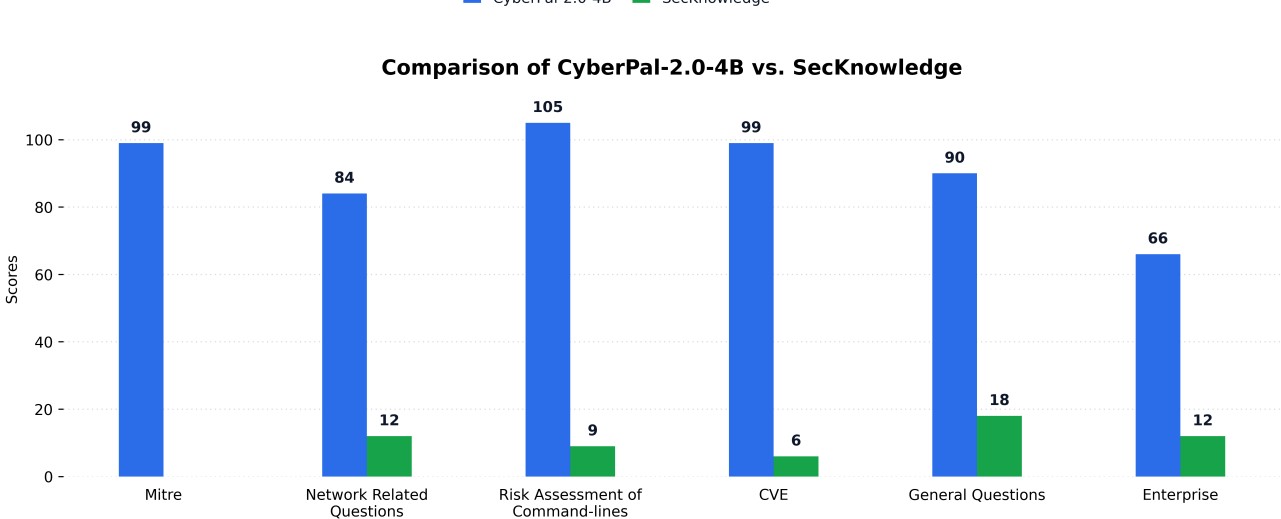

*Figure 12.* LLM-as-Judge pairwise comparison per category: *CyberPal-2.0-4B* vs. *SecKnowledge*.

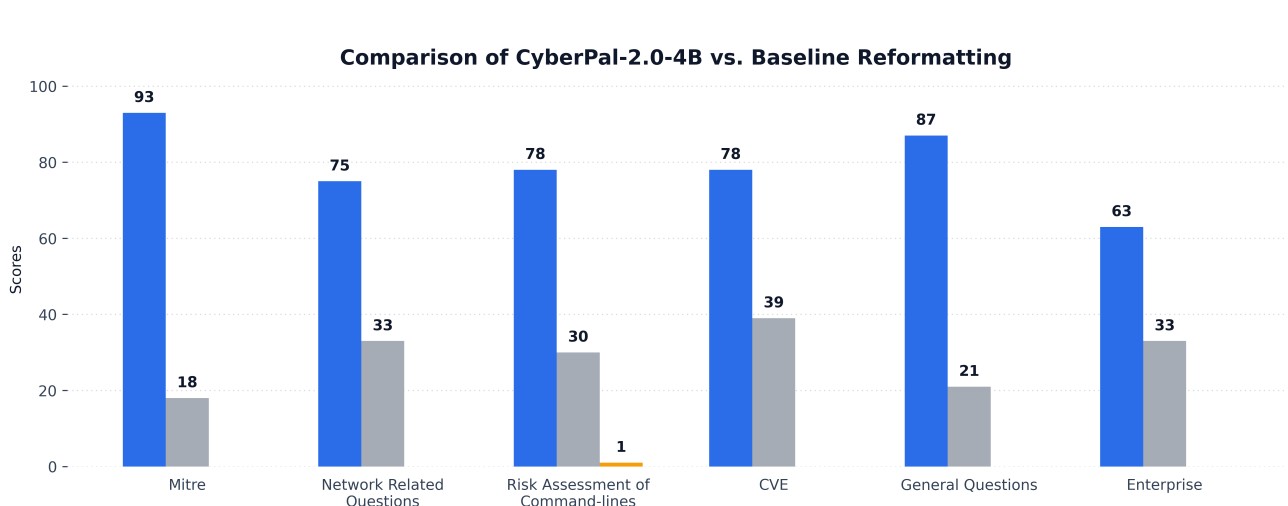

*Figure 13.* LLM-as-Judge pairwise comparison per category: *CyberPal-2.0-4B* vs. *Baseline Reformatting*.

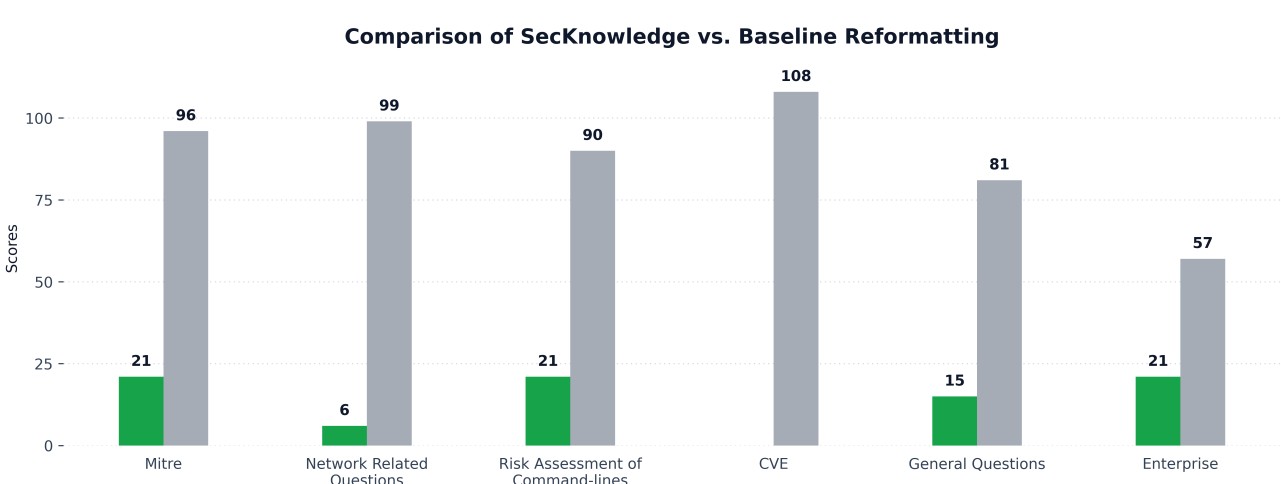

*Figure 14.* LLM-as-Judge pairwise comparison per category: *SecKnowledge* vs. *Baseline Reformatting*.

---

**Instruction**

What are the platforms that are related to the following technique description: Adversaries may make use of Domain Generation Algorithms (DGAs) to dynamically identify a destination domain for command and control traffic rather than relying on a list of static IP addresses or domains. This has the advantage of making it much harder for defenders to block, track, or take over the command and control channel, as there potentially could be thousands of domains that malware can check for instructions.
DGAs can take the form of apparently random or "gibberish" strings when they construct domain names by generating each letter. Alternatively, some DGAs employ whole words as the unit by concatenating words together instead of letters. Many DGAs are time-based, generating a different domain for each time period. Other incorporate a seed value as well to make predicting future domains more difficult for defenders. Adversaries may use DGAs for the purpose of Fallback Channels. When contact is lost with the primary command and control serrver malware may employ a DGA as a means to reestablishing command and control

**SecKnowledge Answer**

The provided description is related to MITRE technique Dynamic Resolution: Domain Generation Algorithms (T1568.002):
Technique Dynamic Resolution: Domain Generation Algorithms (T1568.0002) is applicable to platforms: Windows, MacOS, Linux;
Therefore, the given procedure description is related to platforms: Windows, macOS, Linux.

**SecKnowledge 2.0 Answer**

**Task overview**

We need to determine on which operating environments the technique **Dynamic Resolution: Domain generation Algorithms** (MITRE ATT&CK ID T1568.002) can be executed, based on the provided description.

**Key Indicators In the Description**
- The adversary generates many possible C2 domains rather than using static addresses
- The domains may be random-looking strings or concatenated words, often time-based
- The DGA serves as a fallback channel when the primary C2 link fails.
These characteristics directly match the definition of the sub-technique **Dynamic Resolution: Domain Generation Algorithms.**

**Mapping to supported Platforms**
Based on the described indicators - this behavior can manifest across all major operating systems.
- Windows
- Linux
- macOS

**Final mapping summary**
Therefore, the given procedure description is associated with the platforms: Windows, macOS and Linux

*Figure 15.* Instruction (green) Q&A example from SecKnowledge (green), and our improved answer (orange).

**SecKnowledge Answer**

**Instruction**
Explain ARP Poisoning

ARP (Address Resolution Protocol) Poisoning is a type of cyber-attack which is used to convert IP address to physical addresses on a network device. The host sends an ARP broadcast on the network, and the recipient computer responds back with its physical address. ARP poisoning is sending fake addresses top the switch so that it can associate the fake addresses with the IP address of a genuine computer on a network and hijack the traffic.

**SecKnowledge 2.0 Answer**

**1. Understanding the Question**
The user asks for an explanation of "ARP Poisoning", a term that refers to a specific network-level attack.

**2. Key concepts Identified**
- Address Resolution Protocol (ARP)
- Mapping of IP addresses to physical (MAC) addresses
- Broadcast ARP requests and replies
- Injection of false ARP information to hijack traffic.

**3. Background on ARP and Its Role**
ARP is the protocol that lets a device on a local network discover the MAC (physical) address that corresponds to a given IP address.
- A host broadcasts an ARP request asking "Who has IP X.X.X.X? Tell Y.Y.Y.Y"
- The device that owns that IP replies with its MAC address.
- Because ARP is **stateless,** any host will accept and cache any ARP reply it sees, even if it did not send the corresponding request.
- Cached entries remain until they expire, and new replies overwrite existing ones.
**4. How ARP Poisoning Works**
ARP poisoning (also called ARP cache poisoning or ARP spoofing) exploits the lack of authentication in ARP:
    1. An attack sned **spoofed ARP replies** onto the LAN, claiming that the attacker's MAC address belong to the IP address of a legitimate machine (often the default gateway).
    2. Switches and other hosts update their ARP caches with this false mapping.
    3. Subsequent traffic destined for the genuine IP is now directed to the attacker's MAC address.
    4. The attacker can then intercept, modify, or drop the traffic, effectively hijacking the communication.
**5. Implications and Risks**
- Enables **Main-in-the-Middle (MITM)** attacks, allowing eavesdropping or data manipulation.
- Can be a stepping stone to **Denial-of-Service** attack if th e attacker simply discards the intercepted packets.
- Affects any network that relies on ARP (most Ethernet LANs)

**6. Potential Mitigations and Defenses**
- **Static ARP entries** for critical hosts (e.g., routers) prevent dynamic updates.
- **Detection software** that monitors ARP tables for sudden changes or duplicate IP-to-MAC mapping (e.g., ArpON, XArp).
- **OC-level hardening**: configuring Windows registry keys, enabling Linux's "arp_ignore" / "arp_announce" setting, or using OpenBDS's permanent ARP entires.
- **Network-device features**: such as DHCP-based ARP certification, switch-level ARP inspection, or MAC-anti-spoofing on NICs.

**7. Reasoning Behind the Explanation**
By first clarifying ARP's normal operation, we expose the exact weakness that the attack leverages (stateless, unauthenticated replies). Detailing the attacker's steps shows how the false mapping is created any why traffic is redirected. Including the consequences and defenses completes the picture, ensuring the answer is comprehensive and actionable.

**8. Illustrative Examples**
Suppose Host A (IP 192.168.1.10) want to send data to the internet via the gateway at 192.168.1.1. An attacker on the same LAN broadcasts an ARP reply: "192.168.1.1 is at MAC AA:BB:CC:DD:EE:FF" (the attacker's MAC). Host A updates its cache, so all future packets to 192.168.1.1 go to the attacker, who can forward them to the real gateway after inspection.

**9. Summary of Reasoning**
- ARP maps IP -> MAC via broadcast request/replies.
- Because ARP trusts any reply, an attacker can inject false mappings.
- This "poisoning" redirects traffic to the attacker, enabling interception or disruption.
- Defenses focus on static mappings, monitoring, and OS/network hardening.

**10. Final Answer**
ARP poisoning is a network attack that exploits the Address Resolution Protocol's lack of authentication. An attacker sends forged ARP replies, causing devices on the LAN to associate the attacker's MAC address with the address of a legitimate host (often the default gateway). AS a result, traffic intended for that legitimate host is rerouted to the attacker, who can then sniff, alter, or block the data. The attack works by corrupting the ARP cache entries that devices store after receiving any ARP reply, and it can be mitigated through static ARP entries, ARP-monitoring tools, and operating-system or switch-level security configurations

*Figure 16.* Instruction (green) Q&A example from SecKnowledge (green), and our improved answer (orange).

