# OpenReview forum: "Toward Cybersecurity-Expert Small Language Models"
_ICML.cc/2026/Conference — ICML 2026 regular_

### Official Review · Reviewer_SYKo · 2026-03-04

**Soundness:** 3
**Presentation:** 3
**Significance:** 3
**Originality:** 3
**Overall Recommendation:** 4
**Confidence:** 4

**Summary:**

This paper addresses the important problem that LLM performance on security-critical tasks is abysmal, and one of the primary reasons is the lack of datasets and realistic benchmarks for testing models before deployment in the real world. This paper builds on prior research on creating a domain-knowledge-driven instruction dataset that incorporates domain expertise through structured automation and expert-in-the-loop curation. They use the dataset to train multiple SLMs for security-focused tasks and show that they outperform current SOTA closed models.

**Compliance With Llm Reviewing Policy:**

Affirmed.

**Final Justification:**

My final recommendation is Weak Accept, based on the paper’s practical contribution to the cybersecurity domain through the creation of a high-quality instructional dataset and a suite of specialized SLMs. The work addresses a critical gap in dataset scarcity and provides a robust expert-in-the-loop workflow that ensures domain expertise is effectively distilled into reasoning traces. I found the inclusion of LLM-guided search and document-grounding pipelines to be a significant strength, as it directly mitigates the risk of hallucination in security-critical tasks.

While the authors’ rebuttal provided helpful clarification on the role of LLMaaJ, specifically that it functions as a signal for style and factuality rather than a standalone domain expert, I remain somewhat cautious regarding the 90% alignment figure. My concerns about the potential for misalignment between LLM judges and domain experts in high-stakes security contexts are not fully resolved; however, the authors' validation against human experts and reference to expanded literature provide a reasonable baseline for the current scope. Ultimately, the paper’s strong originality and its clear "engineering direction" outweigh these limitations. I believe the work offers a technically solid foundation that the community will find highly valuable for future cybersecurity AI research.

**Key Questions For Authors:**

- For " EXPERT-IN-THE-LOOP SYSTEM FOR AUTOMATING DOMAIN-SPECIFIC FORMATS", what is the purpose of the auxiliary feedback received from LLM-as-a-judge since the expert is going to edit the format based on their understanding? The LLM that you employed to give the concise task description and instruction-response example, did you test it on other cybersecurity benchmarks to select the best-performing one?
- In the "Training Recipe", you sample approximately 25% of the original instructions, focusing primarily on short, high-quality examples selected using LLMaaJ, but how did you ensure that the LLMaaJ is a good judge to begin with?
- You mentioned "To validate the judge, we measured agreement with human experts and found that, with proper grounding, o3 aligns with human preferences in over 90% of cases." is 90% a good number? Which cases do not align? Why did you use  OpenAI’s o3 as the judge?

**Limitations:**

- Using  LLM-as-a-Judge (LLMaaJ) to assess answer quality is the biggest limitation, which needs to be discussed extensively.

**Strengths And Weaknesses:**

Stengths
+ The paper indeed touches upon the significant dataset scarcity problem in the security domain
+ The cybersecurity instruction dataset reformats raw answers into CoT reasoning traces, encouraging systematic reasoning, which enables their SLM to be fine-tuned precisely on the cybersecurity tasks
+ Expert-in-the-loop workflow ensures that domain-specific knowledge is distilled into reasoning traces that will benefit the LMs later that will be trained on the cybersecurity instruction dataset
+ The work releases a suite of cybersecurity-expert SLMs since frontier Language Models(LM) do not fare well on core cyber threat intelligence tasks, while being closed source, costly, and requiring cloud deployment (privacy-sensitive data are required to leave the enterprise and be stored in a third-party storage location that uses it to train their LM)
+ LLM-GUIDED SEARCH AND DOCUMENT GROUNDING PIPELINES is one of the most critical components since evidence retrival and  grounding [1] ensures that hallucinated and underconfident responses are properly augmented

Weakness:
- The author mentions that "... SecKnowledge increased both diversity and difficulty through dynamic content-grounded synthetic generation...", but how did the author ensure that the synthetic responses are correct and realistic?
- Assessed answer quality via LLM-as-a-Judge (LLMaaJ), this is extremely risky since you are assuming that the judge model is trained properly, but in the beginning, we established that current LLMs are not sufficiently trained for security-critical tasks.

Refs:
[1] LLM-driven Provenance Forensics for Threat Investigation and Detection

---

> ### Author Rebuttal · Authors · 2026-03-31
>
> > **Weakness 1:** How did the author ensure that the synthetic responses are correct and realistic?
>
> We do not rely on unconstrained synthetic generation. The pipeline is designed to preserve correctness through three mechanisms: (i) expert-in-the-loop format design, where experts can inspect rewritten outputs and iteratively refine the format; (ii) explicit grounding through attached documents or targeted retrieval; and (iii) post-generation quality assessment for readability and factuality. For factuality in particular, the rewritten answer is compared against the original answer, which serves as the ground-truth source in this stage. Together, these components were intended to keep the enriched responses both realistic and faithful to the original underlying information.
>
> > **Weaknesses 2 & Questions 2–3**: LLM-as-a-Judge reliability and validation
>
> We agree that LLMaaJ in a security setting must be used carefully. Critically, our judge is not acting as an independent cybersecurity expert - it plays two narrow roles only: (i) a style-preference signal comparing original vs. rewritten answers for readability, and (ii) a factuality check assessing whether the rewrite preserves the content of the original. This turns the problem from open-ended domain judgment into a classic natural language comparison task, where LLMs are typically much more reliable. See Appendix A.2 for a more detailed description of (i) and (ii), and Figure 9 for the LLMaaJ results on different tasks in SecKnowledge 2.0.
>
> To validate this, we used o3 (the most capable model available to us at the time) and obtained >90% agreement with domain experts. Prior work reports human-human agreement of ~80% in similar settings [1], and domain-specific studies report 82–86% LLM-human alignment [2,3] — so we view our result as strong evidence of reliability. We did not use o3 for the full pipeline due to cost; instead we used gpt-oss-120B, a more practical choice for dataset-scale generation while still offering strong reasoning capabilities.
>
> > **Question 1a:** What is the purpose of the auxiliary feedback received from LLMaaJ?
>
> The purpose of the auxiliary LLMaaJ feedback is to accelerate expert iteration, not to replace expert judgment. More broadly, the goal of this interface is not simply to have experts assess LLM outputs, but to support a human-in-the-loop workflow for scalable supervision design. Instead of allowing the LLM to define the template on its own, **experts design and iteratively refine the format alongside the LLM**, so the final template reflects both expert preferences and expert approval. In our interface, experts can inspect rewritten outputs, retrieved evidence, and judge scores side by side while refining a format. In practice, this provides a quick signal when a candidate format is failing on readability or factual consistency, and helps experts prioritize which templates need further revision. The final authority remains with the expert, who can identify domain-specific issues not captured by the judge and manually adjust the format accordingly.
>
>
> > **Question 1b:** The LLM that you employed to give the concise task description and instruction-response example, did you test it on other cybersecurity benchmarks to select the best-performing one?
>
> We apologize if this part was not sufficiently clear in the paper - The concise task descriptions were not selected by benchmarking LLMs on external cybersecurity tasks. In practice, these descriptions were usually written manually by cybersecurity experts** and are intentionally very short.
>
> The instruction–response examples were also not generated by a LLM; they were typically taken directly from the original SecKnowledge seed dataset and then used inside the iterative format-design loop. In other words, the LLM was used as an assistant during template refinement, not as the source of task definitions or domain expertise.
>
> For a more detailed description of this process, we refer the reviewer to Appendix A.1, and to Figure 8, which shows the annotator interface and the information available during format refinement.
>
>
> **References:**
>
> [1] Judging LLM-as-a-Judge with MT-Bench and Chatbot Arena (Zheng et al., 2023)
>
> [2] LLM-as-a-Judge for Low-Resource Languages: Adapting Ragas and Comparative Ranking for Romanian (Creanga et al., 2026)
>
> [3] LLMs as Medical Safety Judges: Evaluating Alignment with Human Annotation in Patient-Facing QA (Diekmann et al., 2025)
>
> ---
>
> We hope our responses and additional results answered any concern the reviewer had, and we hope that the reviewer may consider raising our score.
>
> Best regards, Authors.

---

> > ### Author Rebuttal · Reviewer_SYKo · 2026-04-01
> >
> > I thank the authors for their response, but I still have concerns about the alignment between LLMaaJ and Domain expert. I maintain my score because of the paper's engineering direction.

---

### Official Review · Reviewer_cmA6 · 2026-03-06

**Soundness:** 2
**Presentation:** 1
**Significance:** 1
**Originality:** 1
**Overall Recommendation:** 2
**Confidence:** 5

**Summary:**

This paper presents CyberPal 2.0, a family of cybersecurity-focused small language models (4B–20B parameters) trained on SecKnowledge 2.0, an enriched instruction dataset built through expert-in-the-loop format definition, LLM-driven search and document grounding, and chain-of-thought reformatting. The models are evaluated on nine cybersecurity benchmarks and shown to outperform baselines by 7–14% on average, with the 20B model surpassing several frontier models on root-cause mapping tasks.

**Compliance With Llm Reviewing Policy:**

Affirmed.

**Final Justification:**

After reading all reviews, rebuttals, and follow-up replies, the core issues remain: gains are inseparable from 120B-teacher distillation, source-level contamination between training data and benchmarks is unaddressed, and the contribution combines established techniques (CoT reformatting, RAG, expert annotation) without sufficient methodological novelty for ICML.

Hence, I maintain **my rejection**.

**Key Questions For Authors:**

see the weaknes

**Strengths And Weaknesses:**

## Strengths

1. The paper provides an unusually thorough evaluation suite covering nine cybersecurity benchmarks across diverse security domains (CTI, governance, adversarial robustness, etc.), with detailed taxonomy-based coverage analysis in the appendix.

2. The practical deployment considerations are well-addressed, including quantization experiments (Appendix I) and inference latency measurements (Appendix L), which are useful for practitioners considering real-world adoption.

## Weaknesses

1. **The data enrichment pipeline is essentially distillation from a much larger model, making the claimed "small model" advantage misleading.** The entire SecKnowledge 2.0 dataset is generated/reformatted using "gpt-oss-120B with Medium reasoning effort" as the backbone LLM. The improvements attributed to CyberPal 2.0 are therefore largely a function of distilling knowledge from a 120B-parameter teacher into smaller students. The paper does not adequately disentangle the contribution of the pipeline design (expert-in-the-loop formats, search grounding) from the straightforward effect of having a stronger teacher model rewrite all the training data. The backbone LLM ablation (Table 8, Appendix G.2) only swaps between two large models (gpt-oss-120B vs. Llama Maverick) and shows nearly identical average performance (72.68% vs. 72.32%), suggesting the pipeline is relatively insensitive to the specific teacher — which undermines the claim that the expert-driven format design is the key driver.

2. **The comparison with frontier models is methodologically inconsistent and cherry-picked.** The frontier model comparison (Figure 1) is limited to only two benchmarks (CTIBench-MCQ and CTIBench-RCM), following the Sec-Gemini v1 evaluation protocol. However, several of the other seven benchmarks (e.g., SecEval, CyberMetric-2000) are not reported for frontier models, making it impossible to assess whether CyberPal 2.0 truly matches frontier performance broadly or only on selected tasks. Furthermore, the CyberPal models were specifically fine-tuned on cybersecurity data that likely overlaps in domain coverage with these benchmarks, while frontier models are general-purpose. This comparison is inherently favorable to the specialized model and does not demonstrate a genuine architectural or methodological advance.

3. **The training data likely contaminates several evaluation benchmarks, and no contamination analysis is provided.** SecKnowledge is built from public security sources including MITRE ATT&CK, CWE, CAPEC, BRON, and Sigma rules. Many of the evaluation benchmarks (CTIBench-MCQ, CTIBench-RCM, Adversarial CTI, CTI Detection & Mitigation, CTI Relationship Prediction, Technical Weakness Impact Mapping) are also derived from these same MITRE/CWE/CVE sources. The paper does not perform any train-test overlap analysis or contamination check. The large gains on CTIBench-RCM (+16–31% over baselines) could partly reflect memorization of CWE-CVE mappings present in the training data rather than genuine reasoning capability. This is a significant oversight for a paper claiming frontier-level performance.

4. **The paper is primarily an engineering contribution with limited methodological novelty for ICML.** The core method is: (1) use an LLM to rewrite training data with CoT formatting, (2) add search-based grounding, and (3) have experts define output schemas. Each of these components is well-established — CoT data reformatting follows ReAlign (Fan et al., 2024), search-grounded generation is standard RAG, and expert-in-the-loop annotation is common practice. The combination is sensible engineering but does not introduce new algorithms, architectures, training objectives, or theoretical insights. The paper reads more as a system/application paper suitable for a security or NLP applications venue than a contribution to machine learning methodology.

---

> ### Author Rebuttal · Authors · 2026-03-31
>
> > **Weakness 1:** The data enrichment pipeline is essentially distillation from a much larger model, the paper does not adequately disentangle the contribution of the pipeline design.
>
> We thank the reviewer for raising this point. We agree it is important to separate teacher assistance from pipeline design, and believe our current ablations already partially do so. In both the baseline reformatting condition and our full pipeline, a stronger LLM is used to rewrite the training data, so both settings benefit from teacher assistance; the key difference is therefore the design of the pipeline. **As shown in Figure 6 and detailed in Appendix G.1, our pipeline consistently outperforms the baseline reformatting variant**, suggesting that the gains come from the cybersecurity-specific design of the pipeline rather than from teacher assistance alone. Appendix G.3 further supports this conclusion by isolating the retreival component and showing that removing retrieval hurts performance.
>
> Together, these ablations indicate that the gains come from the pipeline components - especially expert-defined task formats and grounding - rather than simply using a stronger model to rewrite the data. The LLM serves as a constrained rewriter over the original answer, format, and evidence, rather than a source of new cybersecurity knowledge.
> > **Weakness 2:** Comparison with frontier models is inconsistent and not fair
>
> We agree that comparisons between frontier general-purpose models and our cybersecurity-specialized models should be interpreted with care. Our goal was not to claim a fully like-for-like comparison, but to show that much smaller domain-specialized models can be competitive with much larger frontier systems on important cybersecurity tasks.
>
> This is a central point of the paper: CyberPal 2.0 is substantially smaller and more cost-efficient to deploy than frontier models while still achieving competitive results. To our knowledge, prior cybersecurity-LLM literature consistently adopts CTIBench-MCQ/RCM for this type of comparison [1-4]. We therefore focused on these two benchmarks in line with established practice, rather than cherry-picking favorable results.
>
> We also refer the reviewer to Appendix E, where we compare CyberPal 2.0 against other open-source security-specialized models, providing a more direct apples-to-apples comparison.
>
> To address the reviewer’s concern more directly, we additionally report below frontier-model results on the other benchmarks in our evaluation suite, showing that our models are competitive beyond the two benchmarks highlighted in the main paper.
>
> | Benchmark | GPT-4o | o1 | o3-mini | CyberPal-2.0-14B |
> |---|---|---|---|---|
> | CTI Bench MCQ | 74.84 | 68.32 | 67.20 | **75.51** |
> | CTI Bench RCM | 74.19 | 71.26 | 69.94 | **86.00** |
> | SecEval | 64.44 | 49.63 | 68.56 | **69.71** |
> | Cyber Metric 2000 | **93.45** | 91.20 | 91.60 | 89.95 |
> | CISSP Exams | **90.91** | 89.90 | 87.37 | 90.40 |
> | Adv. CTI | **89.70** | 89.14 | 70.24 | 89.58 |
> | Weakness Impact Mapping | **75.45** | 69.46 | **75.45** | 70.77 |
> | CTI Detect & Mitigate | **72.20** | 61.41 | 63.07 | 70.95 |
> | CTI Relationship Prediction | 71.72 | 72.75 | 63.88 | **92.93** |
> | Average | 78.54 | 73.67 | 73.03 | **81.76** |
> > **Weakness 3:** Potential data contamination
>
> We appreciate the reviewers for raising the concern about potential data contamination. Due to character limitation in each response, **please see the answer to the first question of reviewer `JNFz`.**
> > **Weakness 4:** Novelty of contribution
>
> We thank the reviewer for this thoughtful comment. We believe the contribution goes beyond an engineering combination of existing components. At a high level, the paper presents a scalable framework for converting a heterogeneous dataset, with limited expert effort, into a higher-quality version with more structured and grounded supervision. A small group of experts defines and iteratively refines task-specific formats through an expert-LLM loop, and the system then uses these formats to rewrite the dataset at scale.
>
> Thus, the contribution is a general method for supervision improvement, rather than a one-off pipeline for a single benchmark. As shown in Figure 6, this improved supervision leads to consistent gains over standard baseline reformatting.
>
> In addition, we recently open-sourced both the [SecKnowledge 2.0 pipeline](https://github.com/d4nieldev/fms-dgt/tree/secknowledge2/fms_dgt/public/databuilders/secknowledge2) and the [CyberPal 2.0-20B](https://huggingface.co/cyber-pal-security/CyberPal-2.0-20B) to encourage broader adoption and support further progress in the community.
>
>
> **References:**
>
> [1] https://arxiv.org/abs/2601.22159
>
> [2] https://arxiv.org/abs/2601.22975
>
> [3]  https://arxiv.org/abs/2502.11191
>
> [4] https://arxiv.org/abs/2504.21039
>
> ---
>
> We hope our responses and additional results answered any concern the reviewer had, and we hope that the reviewer may consider raising our score.
>
> Best regards, Authors.

---

> > ### Author Rebuttal · Reviewer_cmA6 · 2026-04-01
> >
> > # Maintain Score 2 (Reject)
> >
> > Thank you for the thougtful rebutal and the additional experiments. I appreciate the effort, but unfortunatley my major concerns remain unresolved and I'm maintaining my **score of 2 (reject)**.
> >
> > On the distillation concern, the argument remains ciruclar. Both the baseline and the proposed pipeline use the same teacher LLM, so Figure 6 cannot simpley isolate the contribution of expert-defined formats from teacher assistance. Pointing to the same figure that originally raised the concern is not sufficient.
> >
> > On the frontier comparison, the new table actually reveals that CyberPal underperforms GPT-4o on several benchmarks (CyberMetric, CISSP, Weakness Impact Mapping). This should be acknowleged more openly rather than selectively highlighting favorable results.
> >
> > On contamination, the n-gram and cosine similarity analysis misses the deeper issue: both the training data and benchmarks are derived from the same upstream sources (MITRE ATT&CK, CWE, CAPEC). No surface-level filter can capture this source-level overlap. Reviewer JNFz raised a similiar concern and I dont think it is resolved either.
> >
> > On novelty, generalizability to other domains (beyond cybersecurity) remains an unsupported assertion. The authors argue this is a "general framework" but this claim is unsupported by any evidence beyond cybersecurity. At its core this is a solid engineering paper, and I think it deserves to be published, just not at ICML.
> >
> > Finally, **I would kindly remind the authors that explicitly urging reviewers to raise their scores at the end of each response is considered inappropriate in peer review. Score changes should be left to the reviewers' own judgment.**

---

> > > ### Author Response · Authors · 2026-04-01
> > >
> > > We thank the reviewer for the follow-up and fully respect their assessment. We only clarify several points.
> > >
> > > **Distillation.** We agree that further isolating the effect of the reformatting model would be useful. At the same time, replacing it with a smaller model would not isolate only teacher assistance, since it would also change the model’s ability to handle the long, noisy retrieved context used by our pipeline. This is particularly relevant because the reformatter must jointly process the instruction, original answer, task format, and retrieved evidence. Our current ablations therefore isolate the pipeline more directly: Appendix G.1 compares our expert-guided pipeline to baseline reformatting under the same strong reformatter, and Appendix G.3 shows an additional drop when search is removed.
> > >
> > > **Frontier comparisons.** We do not claim that CyberPal outperforms frontier models on every benchmark. Our frontier-model comparison followed the same industry-standard CTIBench evaluation protocol used by systems such as Sec-Gemini v1 [1], rather than selectively reporting only favorable cases. On these benchmarks, our models are competitive with frontier systems on core cybersecurity tasks. More broadly, in the additional comparison table, CyberPal-2.0-14B achieves the best average result overall, ranks first on 4 of 9 benchmarks, and remains competitive on the others despite being far smaller than proprietary frontier systems. In the final revision, we will revise the framing to emphasize competitiveness rather than broad superiority claims.
> > >
> > > **Contamination.** We agree that contamination should be treated carefully. We therefore clarify that for the internal benchmarks (CISSP Exams, Adv. CTI, Weakness Impact Mapping, CTI Detect & Mitigate, CTI Relationship Prediction), the splits were constructed at the document level and subject level, making benchmark-instance contamination there effectively impossible.
> > >
> > > **References**
> > > [1] https://security.googleblog.com/2025/04/google-launches-sec-gemini-v1-new.html

---

### Official Review · Reviewer_Fb5N · 2026-03-13

**Soundness:** 3
**Presentation:** 2
**Significance:** 3
**Originality:** 2
**Overall Recommendation:** 4
**Confidence:** 5

**Summary:**

The paper introduces a dataset enrichment pipeline that integrates domain knowledge to improve reasoning traces for security-related tasks. Using this pipeline, the authors construct a dataset and train cybersecurity-oriented SLMs. Experimental results demonstrate that these models consistently achieve better performance than both baseline models and almost all tested frontier models.

**Compliance With Llm Reviewing Policy:**

Affirmed.

**Final Justification:**

Although the main novelty appears limited, the paper has a clear strength in that the proposed SLMs outperform state-of-the-art open-source models on CTI benchmarks. In addition, the authors have addressed my main concern regarding the possibility of data contamination well. Therefore, I will raise my score.

**Key Questions For Authors:**

- Was any data leakage analysis conducted between SecKnowledge 2.0 and the evaluation benchmarks? Although benchmark contamination from the base model’s pretraining corpus is always possible, overlap introduced through the fine-tuning dataset is a separate concern and seems more directly attributable in this setting. I would also have liked to see an ablation study on SecKnowledge-Eval.

- It's a bit unclear whether the reported gains come from the proposed data reformatting and enrichment pipeline, or simply from exposing the model to a larger number of evidence/documents during finetuning. For example, consider an ablation that uses only the original grounding documents without additional formatting.

**Limitations:**

Yes

**Strengths And Weaknesses:**

Strengths
- The proposed SLMs outperform both baseline models and state-of-the-art open-source models on CTI benchmarks.

Weakness
- The main novelty seems to lie more in the data than in the modeling approach, but the dataset itself is also an extension of an existing dataset, which kind of limits the novelty aspect.
- The ablation study could be a bit more comprehensive.
- The figures appear in a somewhat confusing order, and this makes the paper a bit harder to follow.

---

> ### Author Rebuttal · Authors · 2026-03-31
>
> > **Weakness 1:** The main novelty seems to lie more in the data than in the modeling approach, but the dataset itself is also an extension of an existing dataset, which kind of limits the novelty aspect.
>
> We agree that the novelty is primarily methodological and data-centric. The paper introduces a framework for scalable supervision improvement in expert domains: we show how a small group of domain experts can define (and iteratively refine) formats that encode how each task should be decomposed and answered, and then use those formats to automatically transform a large heterogeneous dataset into better supervision. In our setting, this produces a clear empirical gain over the original SecKnowledge baseline (+4.08% on average) and over standard baseline reformatting (+2.64% on average). While we evaluate the framework only in cybersecurity in this work, we believe the underlying idea - using domain expert knowledge, experience, and preferences to shape structured supervision at scale - may also be relevant to other complex domains where expert knowledge is needed.
>
> > **Weaknesses 2 and 3:** The ablation study could be a bit more comprehensive, and the figures appear in a somewhat confusing order, and this makes the paper a bit harder to follow.
>
> The paper already includes several ablation studies, though they are currently spread across the main paper and appendix:
>
> 1. **Original SecKnowledge vs. SecKnowledge 2.0** - shown in **Figure 6**
> 2. **Full pipeline vs. baseline reformatting pipeline** - shown in **Figure 6**, with detailed results in **Appendix G.1**
> 3. **Replacing the backbone reformatting LLM** - reported in **Appendix G.2**
> 4. **Removing the search / retrieval component** - reported in **Appendix G.3**
> 5. **Fine-tuning from base vs. post-trained checkpoints** - reported in **Appendix B**
>
> We agree, however, that the current presentation makes these ablations harder to track than necessary, in part because some of them were moved to the appendix due to page-limit constraints. In the final revision, we will consolidate them into a single unified ablation section so that the effect of each component is easier to follow.
>
> > **Question 1:** Was any data leakage analysis conducted between SecKnowledge 2.0 and the evaluation benchmarks? Although benchmark contamination from the base model’s pretraining corpus is always possible, overlap introduced through the fine-tuning dataset is a separate concern and seems more directly attributable in this setting. I would also have liked to see an ablation study on SecKnowledge-Eval.
>
> We appreciate the reviewers for raising the concern about potential data contamination. Due to character limitation in each response, **please see the answer to the first question of reviewer `JNFz`.**
>
> > **Question 2:** It's a bit unclear whether the reported gains come from the proposed data reformatting and enrichment pipeline, or simply from exposing the model to a larger number of evidence/documents during finetuning. For example, consider an ablation that uses only the original grounding documents without additional formatting.
>
> We agree this is a meaningful ablation. The current paper already partially addresses this concern in two ways (see Figure 6):
>
> - First, the comparison against the original SecKnowledge isolates the impact of moving from the original raw dataset to the reformatted one.
> - Second, the comparison against the baseline reformatting pipeline (non-expert-defined general templates, data is being generated given the same evidence) isolates the effect of our cybersecurity-specific pipeline relative to a more standard reformatting approach.
>
> Together, these results suggest that the gains are not solely due to exposing the model to more evidence, but also to how the data are structured and grounded.
>
> That said, we agree that a cleaner ablation using only the original grounding documents without the new formatting would be interesting. We were not able to add it during the rebuttal period because it would require rebuilding a large training set with compressed grounding for roughly 600K instructions and then retraining under a larger-context setup, which was beyond the available time.
>
> ---
>
> We hope our responses and additional results answered any concern the reviewer had, and we hope that the reviewer may consider raising our score.
> Best regards, Authors.

---

> > ### Author Rebuttal · Reviewer_Fb5N · 2026-04-03
> >
> > Thank you for addressing my concerns. However, I am still not convinced that the paper demonstrates sufficient novelty for ICML, and I also remain uncertain about the possibility of data contamination. Therefore, I maintain my score.

---

> > > ### Author Response · Authors · 2026-04-05
> > >
> > > We thank the reviewer for the follow-up and for taking the time to reconsider the paper. We agree that contamination should be handled conservatively, and we appreciate the reviewer’s point that pointwise overlap checks alone cannot fully exclude all forms of benchmark similarity.
> > >
> > > One important clarification is that **5 of the 9 benchmarks in our evaluation suite - CISSP Exams, Adv. CTI, Weakness Impact Mapping, CTI Detect & Mitigate, and CTI Relationship Prediction** - are internal benchmarks whose train/test splits were defined at the document and subject level. For these benchmarks, benchmark-instance contamination is therefore effectively ruled out by construction. Since these five benchmarks make up the majority of our evaluation suite, this concern does not apply to most of the reported results.
> > >
> > > More generally, we made every effort to maximize separation between the training data and the evaluation suite, and we hope this clarification helps reduce the reviewer’s concern regarding contamination.

---

### Official Review · Reviewer_JNFz · 2026-03-18

**Soundness:** 4
**Presentation:** 3
**Significance:** 2
**Originality:** 2
**Overall Recommendation:** 4
**Confidence:** 3

**Summary:**

The authors present Small Language Model called 'CyberPal 2.0' with 4B-20B parameters and `Qwen3-{4,8,14}B-base`/`gpt-oss-20B` trained on a pipeline that enriches the existing SecKnowledge cybersecurity dataset. Using a pipeline inspired by ReAlign plus evidence grounding the  cyber evidences come from MITRE, CVEs, etc. and web search results. Cyber reasoning competency on the fine-tuned 'CyberPal 2.0' is evaluated against nine sophisticated cybersecurity benchmarks containing collections of cybersecurity questions.

Ablation study of the 3 versions of training data -- `SecKnowledge`; `SecKnowledge` w/ vanilla ReAlign; `SecKnowledge` w/ expert-in-the-loop formats and  cyber evidence grounding demonstrates  highest scores wtih 'CyberPal 2.0`. The authors ultimately argue a SLM if fine-tuned and structured with optimal cyber enhanced structure can outperform SoTA models against Cybersecuity knowledege benchmarks.

**Compliance With Llm Reviewing Policy:**

Affirmed.

**Final Justification:**

We thank the authors for their rebuttal and clarifications. The paper has clear strengths -- particularly its strong results for SLM and thoughtful ablation study.

The overall concern is to whether the reported gains reflect actual reasoning or just recall. While the n-gram and cosine checks works towards this resolution, they are not convincing enough to eliminate overall concerns on this front.  For this reason our score remains unchanged.

The authors' response to WQ2 moving toward agentic evaluation is meaningful and a positive direction, with such updates enabling a more convincing case away from memorization.

We encourage the authors to pursue such work in the future, and congratulate them on their strong results.

**Key Questions For Authors:**

1. What contamination analysis could be performed to separate memorization from reasoning in these results?

2. Could applied experiments such as tool-assisted investigations or CTF challenges better demonstrate that the SLM is reasoning rather than recalling?

**Limitations:**

yes

**Strengths And Weaknesses:**

STRENGTHS::

The results are impressive for a SLM, making it competitive on standard cyber benchmarks with LLM of orders of magnitude larger. The training methodolgy, data curation for cyber knowledge retrieval, and process to improve on `SecKnowledge` are well motivated, with ablation study results and final findings showing the success of such a pipeline.

The experiments range to include nine reputable evaluation sets, include the CISSP exam, allow us to assume coverage for wide array of cybersecurity topics.


WEAKNESSES::

The largest weakness with the experiment is that beyond J.3. wider data contamination is largely unaddressed. It is not clear if the SLM are genuinely reasoning, or just well trained on very similar datasets with great similarity to the benchmarks. No diversity analysis was done on test and training to either prove or disprove this possibility.

The study is sound and show promise, and would have benefited greatly from experimentation that demonstrated real world evaluation of security use cases.

---

> ### Author Rebuttal · Authors · 2026-03-31
>
> > **Weakness/Question 1**: Data contamination is not sufficiently addressed, leaving open whether the reported gains reflect genuine reasoning or memorization of training data that is highly similar to the benchmarks; stronger contamination analysis is needed to separate the two.
>
> We appreciate the reviewers for raising the concern about potential data contamination. To assess this, we screened the training data against the evaluation benchmarks using both n-gram overlap and semantic similarity.  We compared every question in the benchmark test sets against every question in the training dataset using two methods: (1) n-gram overlap - checking if long exact word sequences (8-grams) appear in both, and (2) embedding similarity - encoding both questions into vectors and measuring how semantically close they are (cosine similarity). A training sample is flagged as suspicious if it shares more than 50% n-gram overlap, and has cosine similarity above 0.8 with a benchmark question.
>
> This process identified approximately 700 suspicious examples out of ~600K training instructions (which is 0.1% of the train set, and 6.34% of the combined evaluation sets). We then removed these examples and retrained the 4B model under the same training setting. This retraining produced no statistically meaningful change in performance (as detailed in the table below), suggesting that benchmark contamination is unlikely to affect our conclusions.
>
> | Benchmark                     | CyberPal-2.0-4B | CyberPal-2.0-4B (Filtered) | Difference |
> |------------------------------|----------------:|---------------------------:|-----------:|
> | CTI Bench MCQ                | **69.70%**      | 68.92%                     | -0.78%     |
> | CTI Bench RCM                | **81.15%**      | 77.13%                     | -4.02%     |
> | SecEval                      | **59.02%**      | 56.87%                     | -2.15%     |
> | Cyber Metric 2000            | **87.80%**      | 87.60%                     | -0.20%     |
> | CISSP Exams                  | 80.80%          | **82.32%**                 | +1.52%     |
> | Adv. CTI                     | 68.03%          | **68.69%**                 | +0.66%     |
> | Weakness Impact Mapping      | 66.48%          | **74.25%**                 | +7.77%     |
> | CTI Detect & Mitigate        | **64.03%**      | 60.27%                     | -3.76%     |
> | CTI Relationship Prediction  | 77.12%          | **81.36%**                 | +4.24%     |
> | Average                      | 72.68%          | **73.04%**                 | +0.36%     |
>
>
> > **Weakness/Question 2:** The study is promising, but it would be strengthened by more realistic security evaluations—such as tool-assisted investigations or CTF-style tasks - to better demonstrate reasoning beyond recall.
>
> We agree that applied evaluations are valuable for separating recall from operational reasoning. While full agentic evaluations such as end-to-end tool-assisted investigations or CTF-style environments were not the central focus of this paper, we do include additional real-world use-case evaluations in **Appendix J**, including threat-report-to-TTP mapping, CyberSocEval ( Malware Analysis and Threat Intelligence Reasoning), CVE reassessment, and adversarial robustness for insecure code generation (CyberSecEval). These were intended to complement the benchmark results with more applied settings. We are also actively working on a more agentic version of the model, where tool-augmented evaluations will be a natural next step.
>
> ---
>
> We hope our responses and additional results answered any concern the reviewer had, and we hope that the reviewer may consider raising our score.
>
> Best regards, Authors.

---

> > ### Author Rebuttal · Reviewer_JNFz · 2026-04-03
> >
> > Thank you for your update.
> >
> > > WQ1:
> >
> >
> > While the effort to mitigate dataset contamination through n-gram and cosine similarity checks are acknowledged, they are both ultimately pointwise checks. Such an approach is not convincing enough to meaningfully eliminate or significantly prove an upper bound on the amount of data contamination possible.
> >
> > Other research efforts in LLMs show that recognition remains high even on word sequence changes.
> >
> > While CISSP and other cybersecurity knowledge datasets are extensive in breadth, success in these realms often convey memorization as these are static benchmarks, and using such datasets without other tasks limits the capacity to evaluate authentic reasoning.
> >
> >
> > > WQ2:
> >
> > It is good that the authors are continuing in the direction of full end to end agentic evaluations.
> >
> > While WQ1 and WQ2 have been fully addressed, due largely to the weaknesses in WQ1 the evaluation score remains unchanged.

---

> > > ### Author Response · Authors · 2026-04-05
> > >
> > > We thank the reviewer for the careful and thoughtful follow-up. We agree that contamination deserves a cautious treatment, and we appreciate the reviewer’s concern that pointwise overlap checks alone cannot fully rule out all forms of benchmark similarity.
> > >
> > > An important clarification is that **5 of the 9 benchmarks in our evaluation suite - CISSP Exams, Adv. CTI, Weakness Impact Mapping, CTI Detect & Mitigate, and CTI Relationship Prediction** - are internal benchmarks whose train/test splits were constructed at the document and subject level. For these benchmarks, benchmark-instance contamination is therefore effectively impossible by design. Since these account for the majority of our evaluation suite, this concern does not apply to most of the reported results.
> > >
> > > More broadly, we made our best effort to maximize separation between the training data and the evaluation suite, and we hope this clarification helps address the reviewer’s concern regarding contamination.

---

### Decision · Program_Chairs · 2026-04-30

**Decision:**

Accept (regular)

**Comment:**

While reviewers appreciated the reported performance gains for small language models, two common themes were had in the discussion: first, nearly all reviewers were concerned about data contamination. Second, there were some concerns about having the right ablations to attribute where performance gains come from.

The latter was largely addressed during the discussion phase. A reviewer had some remaining concerns on the effect of distillation, however, I agree with the authors point that while one can technically separate the effect of distillation, the existing ablation that keeps the the same large backbone and only changes the pipeline does in fact test the individual pipeline effect, and an experiment that extracts out the effect of distillation is orthogonal to this submission.

The former issue on data contamination is the trickier part. The authors did some level of checking during the rebuttal period, which uncovered a small percentage of nearby contamination using very conservative matching criteria. Actual semantic matches may potentially be much higher, which could be addressed by a powerful enough agentic evaluation, but as of yet unknown. Hence, the general agreement amongst the reviewers is that there's not enough evidence to conclude that there is definitely not significant data contamination, but that the authors did provide some (very strict) evidence that it may not be substantial enough to matter. Hence, none feel confident enough to push for the papers acceptance, and weak accept is ultimately the right call for the state and claims of the paper. Given that building useful small models are ultimately much more accessible to the wider research community, I am hedging on the side of recommending the paper for acceptance and hope that the authors include a full agentic evaluation to help asuade the concerns of future readers.